



# An Overview of the Surface Ocean Aerosol Production (SOAP) campaign

Law Cliff S.[1,2], Smith Murray J.[1], Harvey Mike J.[1], Bell Thomas G.[3,4], Cravigan Luke T.[5,6], Elliott Fiona C.[1], Lawson Sarah J.[7], Lizotte Martine[8], Marriner Andrew[1], McGregor John[1], Ristovski Zoran[6], Safi Karl A[9], Saltzman Eric S.[4], Vaattovaara Petri[10], Walker Carolyn F.[1]

[1] National Institute of Water and Atmospheric Research, Wellington, New Zealand
[2] Department of Chemistry, University of Otago, Dunedin, New Zealand
[3] Plymouth Marine Laboratory, Prospect Place, The Hoe, Plymouth, UK
[4] Department of Earth System Science, University of California, Irvine, CA, USA
[5] Climate Science Centre, Commonwealth Scientific and Industrial Research Organisation, Aspendale, Australia
[6] International Laboratory for Air Quality and Health, Queensland University of Technology, Brisbane, Australia
[7] Commonwealth Scientific and Industrial Research Organisation, Oceans and Atmosphere Flagship, Aspendale, Australia
[8] Université Laval, Department of Biology (Québec-Océan), Québec City, Québec, Canada.
[9] National Institute of Water and Atmospheric Research, Hamilton, New Zealand
[10] University of Kuopio, Dept. of Physics, Kuopio, Finland

Correspondence to Cliff Law  cliff.law@niwa.co.nz

**Abstract.** Establishing the relationship between marine boundary layer (MBL) aerosols and surface water biogeochemistry over the remote ocean is required to understand aerosol and cloud production processes, and also represent them accurately in Earth System Models and global climate projections. This was addressed in the SOAP (Surface Ocean Aerosol Production) campaign, which examined air-sea interaction over biologically-productive frontal waters east of New Zealand. This overview details the objectives, regional context, sampling strategy, and provisional findings of a pilot study, PreSOAP, in austral summer 2011, and the following SOAP voyage in late austral summer 2012. Both voyages characterised surface water and MBL composition in three phytoplankton blooms of differing species composition and biogeochemistry, with significant regional correlation observed between chlorophyll-$a$ and DMSsw. Surface seawater dimethylsulfide (DMSsw) and associated air-sea DMS flux showed spatial variation during the SOAP voyage, with maxima of 25 nmol L$^{-1}$ and 100 μmol m$^{-2}$ d$^{-1}$, respectively,



recorded in a dinoflagellate bloom. Inclusion of SOAP data in a regional DMSsw compilation confirmed

that the current climatological mean is an underestimate for this region of the South-west Pacific. Estimation of the DMS gas transfer velocity ($k_{DMS}$) by independent techniques of eddy covariance and gradient flux showed good agreement, although both exhibited periodic deviations from model estimates. Flux anomalies were related to surface warming and sea surface microlayer enrichment, and also reflected the heterogeneous distribution of DMSsw and the associated flux footprint. Other aerosol

precursors measured included the halides and various volatile organic carbon compounds, with the first measurements of the short-lived gases glyoxal and methylglyoxal in pristine Southern Ocean marine air indicating an unidentified local source. The application of a real-time clean-sector, contaminant markers, and a common aerosol inlet facilitated multi-sensor measurement of uncontaminated air. Aerosol characterisation identified variable Aitken mode, and consistent sub-micron sized accumulation

and coarse modes. Sub-micron aerosol mass was dominated by secondary particles containing ammonium sulfate/bisulfate under light winds, with an increase in sea-salt under higher wind-speeds. MBL measurements and chamber experiments identified a significant organic component in primary and secondary aerosols. Comparison of SOAP aerosol number and size distributions reveals an underprediction in GLOMAP-mode aerosol number in clean marine air masses, suggesting a missing

marine aerosol source in the model. The SOAP data will be further examined for evidence of nucleation events, and also for relationships between MBL composition and surface ocean biogeochemistry with the aim of identifying potential proxies for aerosol precursors and production.


## 1.    Introduction

It is recognised that the surface ocean alters the properties of the lower atmosphere, and so atmospheric albedo and climate (McCoy et al., 2015; Seinfeld et al., 2016), via the direct and indirect effects of aerosols (O'Dowd and de Leeuw, 2007). Aerosols are precursors of clouds, which play a major role in the scattering and absorption of incident solar radiation (Carslaw et al., 2013), but the concentration, number and chemical properties of aerosols that act as cloud-condensation nuclei (CCN)

can also influence cloud droplet size and number, and consequently precipitation and cloud albedo (Twomey, 1977). Indeed, cloud formation and properties are sensitive to relatively minor changes in aerosol concentration, particularly in remote regions (Carslaw et al., 2013). This is particularly the case in the Southern Ocean, where natural aerosol sources dominate and where CCN concentrations can range from tens per $cm^3$ in winter to hundreds per $cm^3$ in summer (Andreae and Rosenfeld, 2008),

leading to seasonally variant trends in cloud albedo. However, the relationship between clouds and aerosols derived from natural sources is poorly understood, and represents a major uncertainty in the representation of low-level marine clouds and feedbacks in climate models (Wang et al., 2013; Stephens, 2005). Current models underestimate cloud over the Southern Ocean, particularly south of 55ºS, resulting in excess surface shortwave radiation and a warm bias (Trenberth and Fasullo, 2010; Kay

et al., 2016). This discrepancy is potentially attributable to a variety of factors, chief among which is the limited understanding of aerosol-cloud interaction and cloud water phase, compounded by a lack of regional observations and data to advance satellite retrievals and climate model simulations.

Breaking waves and associated bubble formation are a major source of Primary Marine Aerosol (PMA), supplying most the aerosol mass in the marine boundary layer (MBL) over the remote ocean (Andreae

and Rosenfeld, 2008). This is particularly so in regions that experience high winds and breaking waves (de Leeuw et al., 2014), with PMA contributing only ~10–20% of CCN number concentrations over the remote Pacific Ocean (Blot et al. 2013; Clarke et al. 2013), but up to 55% over the Southern Ocean (McCoy et al. 2015). Although PMA is generally regarded as primarily composed of sea-salt, recent reassessments suggest it is highly enriched in organic matter relative to bulk seawater. Organic material

may in fact dominate submicron aerosol mass (Facchini et al., 2008; O'Dowd et al., 2004), with the Primary Organic Aerosol (POA) being of biogenic origin and including bacteria, carbohydrates, polymers and gels (Facchini et al., 2008; Russell et al., 2010). Although the contribution of POA to the MBL is uncertain, it may be significant over biologically active oceanic regions, as suggested by correlations between organic aerosol content and surface chlorophyll-*a* (Chl-*a*) (O'Dowd et al., 2004). There is also

similarity in the composition of aerosol and surface ocean organics, and organically enriched sub-micron particles have been produced experimentally using surface seawater conditions (Quinn and Bates,





2011). Indeed, the degree of organic enrichment may influence both the type and size of aerosols, as well as properties such as aerosol light scattering and water uptake (Vaishya et al., 2012).

It is well-established that biologically productive regions are characterised by elevated concentrations and emissions of a range of compounds that may influence aerosol production, composition and properties (Meskhidze and Nenes, 2010; Gantt and Meskhidze 2013; de Leeuw et al., 2014). However, the oceanic influence on atmospheric composition is not only attributable to PMAs but also to secondary marine aerosols (SMAs), that are produced during gas-phase reactions of volatile organic compounds (VOCs). Although SMAs have less impact upon aerosol mass they potentially have a large influence on aerosol number (Meskhidze et al., 2011). The biogeochemical origin of SMAs is reflected in their seasonality, with Aitken and accumulation mode aerosol number concentrations dominated by secondary particles in summertime (Clarke et al. 2013; Cravigan et al. 2015). Research into SMAs has primarily focussed on dimethylsulfide (DMS), the primary natural marine source of volatile sulfur, in response to early hypotheses related to its potential role in climate feedback processes (Charlson et al., 1987). The CLAW hypothesis linked the production of the DMS precursor, dimethylsulfoniopropionate (DMSP), by phytoplankton and subsequent DMS emission and oxidation to sulfate aerosol, to CCN formation and changes in cloud cover. Although well-studied, this hypothesis remains unproven and there is a lack of consensus, with a recent review identifying uncertainties regarding the role of DMS in aerosol production in the MBL (Quinn and Bates, 2011). However, there is evidence that DMS may play a role in cloud formation over larger spatial and temporal scales, via entrainment from the free troposphere (Carslaw et al., 2010).

The fundamental tenet of the CLAW hypothesis, of feedback between surface ocean biogeochemistry and climate, may be applicable via a broader spectrum of precursor species. Recent research has shown increasing complexity of potential aerosol source pathways, involving a variety of chemical species, processes and interactions (Vaattovaara et al., 2006). In addition to DMS, a variety of other gaseous aerosol precursors that originate from phytoplankton, bacterial and photochemical sources at the sea surface may undergo physical and chemical transformation to produce new particles in the MBL (Ciuraru et al., 2015). These SMA precursors include volatile organic species, such as carboxylic acids, isoprene, monoterpenes, halocarbons, iodine oxides and iodine (Vaattovaara et al., 2006; Sellegri et al., 2005). A biological source of these SMAs has been inferred from the spatial and temporal correlation between phytoplankton blooms and cloud microphysics (Meskhidze et al., 2009; Meskhidze and Nenes, 2010; Lana et al. 2012). The presence and concentration of SMA precursors in the MBL may be dependent upon plankton abundance and community composition, and consequently their influence on aerosol formation will show spatial and seasonal variability (O'Dowd et al., 2004).



New particle formation may be suppressed by the interaction of aerosol precursors and SMAs with pre-existing aerosol, for example, by absorption of ammonia and gaseous sulfuric acid by coarse mode sea-salt aerosol (SSA; Cainey and Harvey, 2002). Conversely, existing particles may grow via condensation which enhances their CCN capacity (Clarke et al., 2013). It has also been proposed that organic acids combine with sulfuric acid to create the critical nucleus required for aerosol formation (Zhang, 2010;

Almeida *et al.* 2013). However, nucleation events over the open ocean remain elusive (O'Dowd et al., 2010; Chang et al., 2011; Willis et al., 2016), making it difficult to elucidate the primary pathways and reactants, and consequently they are currently regarded as of low significance to marine aerosol formation. Following nucleation, the aerosol distribution is modified by aerosol-aerosol interaction, heterogenous reactions and removal processes, including coagulation and condensation, resulting in the

longest-lived aerosol component being in the accumulation mode (0.06-0.4μm). With such a wide variety of potential precursors and inorganic/organic interactions affecting nucleation and CCN activation, the modelling of aerosols and their indirect influence on cloud radiative properties over the remote ocean presents a major challenge (Seinfeld et al., 2016).

The production and transfer of aerosol precursors from the ocean surface is also dependent upon

physical factors. Exchange across the air-sea interface is primarily controlled by near-surface turbulence, which is dependent on wind and waves. For practical purposes, this is represented by a kinetic factor, the transfer velocity $k$ which is generated with wind-speed parameterisations (Nightingale et al., 2000; Ho et al., 2006). Although wind-speed provides a reasonable broad-scale proxy for kinetic transfer, other factors such as fetch, wave development, wind-wave direction and surfactants also influence $k$, and so

generate variation in gas exchange and deviation from $k$-wind-speed relationships. For example, most k-wind-speed parameterisations do not explicitly capture the solubility effects associated with bubbles (Blomquist et al. 2006), although the COAREG gas transfer model incorporates this factor into a physically-based flux algorithm (Fairall et al., 2003; Fairall et al. 2011). Biogeochemical gradients near or at the ocean surface are also not considered, despite their potential to alter the air-sea exchange of

gases, PMAs and SMAs (Facchini et al., 2008; Calleja et al., 2013).

The Surface Ocean Aerosol Production (SOAP) campaign was initiated with the primary aim of characterising variation of aerosol composition and concomitant marine sources, processes and pathways in the South-west Pacific. SOAP utilised a multi-disciplinary framework, encompassing surface ocean biology and biogeochemistry, transport and air-sea exchange with characterisation of aerosol

number and composition, to establish controls on aerosols and gas exchange. The campaign consisted of two voyages - a pilot study, PreSOAP, which carried out a regional survey and established sampling strategies, and the following SOAP voyage – in biologically productive sub-tropical front waters along



the Chatham Rise, east of New Zealand (see Figure 1). The following paper details the regional context, sampling strategy, environmental conditions and some preliminary results for the SOAP campaign.

## 2.  Regional context

The South-west Pacific has many features in common with the Southern Ocean, as it is characterised by low anthropogenic and terrestrial aerosol loading, long ocean fetch and high wind-speed, making it an optimal location for examining the marine contribution to aerosol production.  One of the more

biologically productive regions lies east of New Zealand, where the Sub-Tropical Front (STF) extends as a tongue of elevated phytoplankton production (Murphy et al., 2001), along 43.0-43.5ºS over the Chatham Rise (see Figure 1a). This arises from the confluence of warmer saline subtropical waters that are relatively deplete in macronutrients, with fresher cooler subantarctic waters containing elevated macronutrients but depleted in iron (see Figure 1b; Boyd et al., 1999). Mixing across the front alleviates

nutrient stress which, combined with a relatively stable water column, promotes primary production (Chiswell et al., 2013). Ocean colour climatologies show a monthly mean Chl-$a$ of 0.6 mg m$^{-3}$, reaching ~ 1 mg m$^{-3}$ over the Chatham Rise in spring (Murphy et al., 2001), and the region is characterised by elevated marine particle export, secondary production and fish stocks (Nodder et al., 2007; Bradford-Grieve et al., 1999). In spring the phytoplankton community composition varies with water mass, with

diatoms dominating the STF, cryptophytes, prasinophytes and dinoflagellates more prevalent in subtropical waters, and photosynthetic nanoflagellates dominating subantarctic waters (Chang and Gall, 1998; Delizo et al., 2007). The STF also supports spatially-extensive coccolithophore blooms (Sadeghi et al., 2012), and is situated on the northern edge of the "Great Calcite Belt" (Balch et al., 2011), a latitudinal band of elevated backscatter attributed to coccolithophore liths. Surface mixed layer

nutrients vary spatially in response to mixing of the water masses and seasonally due to phytoplankton uptake, with the evolution of nutrient stoichiometry and grazing determining the succession and duration of different phytoplankton blooms (Chang and Gall, 1998; Delizo et al., 2007). The STF is characterised by significant gradients in $pCO_2$ associated with phytoplankton blooms, with current global climatologies indicating the region east of New Zealand as a significant carbon sink (>1mol C m$^{-2}$ yr$^{-1}$,

Landschuetzer et al., 2014).

The waters south of New Zealand are characterised by high wind-speeds which drive the disproportionate contribution of this region to global ocean $CO_2$ uptake. Here, wind, waves and currents develop unhindered by land, and strong persistent westerlies act over long fetch to generate large swells that propagate north-east influencing the wave-climate off New Zealand. While this wave energy is



attenuated closer to land in the eastern Chatham Rise, the average wave energy is still 75% of values south of New Zealand where annual mean wave heights exceed 4m. Subantarctic waters south of the Chatham Rise region provided a prime location for a dual tracer release experiment (SAGE; Harvey et al., 2011), aimed at constraining $k$ at high wind-speeds. Comparison of the SAGE $k$-wind-speed parameterisation with those generated in other regions, and using different techniques, showed

generally good agreement (Ho et al., 2006); this may be interpreted as indicating that regional influences on exchange may be less important, supporting the application of a universal wind-speed parameterisation. Nevertheless, other factors, such as wave age, duration and height do influence gas exchange in this region (Smith et al., 2011; Young et al., 2012). The elevated winds also influence the transfer of aerosol precursors, as reflected by a zonal band of elevated sea spray aerosol mass and

water-insoluble organic matter over the Chatham Rise region (Vignati et al., 2010).

Both models and measurements indicate that DMS is a significant contributor to total non-sea salt sulfate (nssSO4) in the Southern Hemisphere (Gondwe et al., 2003; Korhonen et al., 2008). However, a paucity of observational data in the Southern Ocean has hindered development of global climatologies for surface seawater DMS (DMSsw), with the region south-east of New Zealand represented by only a

few data points in a recent DMS climatology (Lana et al., 2011). Despite this shortcoming, this climatology provides a realistic representation of atmospheric DMS and total sulfate when applied in aerosol-climate Global Climate Models, particularly over the Southern Ocean (Mahajan et al., 2006). Seasonal variability in atmospheric DMS is apparent at stations in New Zealand and south of 44°S (Blake et al., 1999), with concentrations of 100-200 pptv, and maximal values associated with the transport of

DMS from waters to the south in summer (Harvey et al., 1993; de Bruyn et al., 2002; Wylie and de Mora, 1996). Corresponding seasonality in nssSO4 was observed, with a maximum (0.8-1.5 $\mu g\ m^{-3}$) in early austral summer at the start of the year, decreasing in late summer to 0.1-0.4 $\mu g\ m^{-3}$ through autumn and winter (see Figure. 2; Sievering et al., 2004; Allen et al., 1997). For comparison, coarse SSA dominates the aerosol mass at Baring Head, with concentrations of 6-10 $\mu g\ m^{-3}$ (Jaeglé et al., 2011;

Spada et al., 2015).  Similar seasonal cycles of DMS and nssSO4 were recorded at Cape Grim (Ayers, 1991), and the observed diurnal inverse correlation between sulfur dioxide and DMS at Baring Head was applied to estimate yield and potential contribution to aerosols (de Bruyn et al., 2002). Consistent seasonal trends between activated particles and cloud droplet number concentration were also apparent, with a summer maximum over the Southern Hemisphere (Boers et al., 1996; 1998), related

to phytoplankton production (Thomas et al., 2010). Overall, the temporal trends in aerosol precursors and pathways do not follow that of wind-speed and other physical drivers, but instead reflect biological processes inferring control by surface ocean biogeochemistry (Korhonen et al., 2008).



### 3. Research Programme and Strategy

**3.1 PreSOAP**

A pilot study, PreSOAP, was carried out to test technical approaches and confirm the regional source of biogenic aerosols in the Chatham Rise region on the New Zealand research vessel, *Tangaroa*, on 1-12/2/2011 (DoY32-42). The strategy of bloom location using satellite imagery and subsequent mapping of surface properties proved successful, with three blooms of differing DMSsw and $pCO_2$ signatures

located and monitored each for 3-4 days. The first bloom was initially dominated by dinoflagellates with an increase in diatom biomass after 3 days, while the second and third blooms were primarily dominated by coccolithophores and dinoflagellates, respectively. This variability in species composition resulted in significant spatial and temporal variability in DMS concentrations in the MBL (DMSa) and DMSsw. DMSa concentration varied over two orders of magnitude, reaching 1000 ppt on DoY 36 (see Figure 3b), similar

in range to that recorded at the Baring Head station near Wellington (Harvey et al., 1993; de Bruyn et al., 2002). There was no significant correlation between DMS in the two phases, with DMSa showing a stronger relationship with wind-speed (see Figure 3). Surface Chl-*a* concentrations reached 2 mg m$^{-3}$, but there was no significant relationship between DMSsw and Chl-*a*, with the DMSsw maximum of ~10 nmol L$^{-1}$ during the first bloom coinciding with Chl-*a* of ~1 mg m$^{-3}$ (Figure 3d). The observed temporal

and spatial variability in DMSa and DMSsw during PreSOAP highlighted the technical challenge of establishing relationships between surface ocean biogeochemistry and atmospheric composition. Provisional method development was also carried out for measurement of DMS and other parameters in near-surface waters and the sea surface microlayer (SSM).

Surface DMSsw and $pCO_2$ were mapped, and DMSa and $CO_2$ MBL concentrations and fluxes measured

continuously by sensors and collectors mounted on the bow of the vessel. Testing of the eddy covariance (EC) flux technique identified an issue with water vapour interference that dominated the $CO_2$ signal recorded by an open-path InfraRed Gas Analyser (IRGA). Preliminary studies also identified that residual ship motion dominated over turbulence for the real-time switching of Relaxed Eddy Accumulation measurement of flux under high swell conditions. The logistical challenges of flux measurement at

distance from the vessel were also assessed by deployment of a free-floating catamaran supporting a mounted gradient flux sampling system (Smith et al., to be submitted). A temperature microstructure profiler was also deployed to record near-surface temperature and turbulence structure (Stevens et al., 2005), although this was limited to short sampling periods, highlighting the need for a mounted thermistor array on a spar buoy for longer measurement coverage.



The utility of a baseline sector for sampling MBL composition, using relative wind direction and speed, was also tested during PreSOAP. Measurements showed a tendency for higher condensation nuclei concentration in the "non-baseline" sector, confirming the utility of this approach (Harvey et al., to be submitted). A common aerosol inlet provided clean air from a height of 17.5m above sea level to instruments and sensors in a container laboratory on deck. Particle size distribution and concentration, including ultrafine nuclei concentrations, were continuously monitored using a scanning mobility particle sizer (SMPS) and optical particle counters (OPC's), with bulk ion chemistry samples collected using a high-volume sampler (see Supplementary Table 1). Experiments examining the source and organic content of aerosols were carried out on surface water samples in a laboratory-based bubble chamber.

## 3.2 The SOAP voyage

The SOAP voyage employed the strategy successfully piloted on PreSOAP, of identifying phytoplankton blooms in NASA MODIS Aqua and Terra satellite ocean colour images, with subsequent bloom location and mapping using a suite of underway sensors (Chl-$a$, $\beta_{660}$ backscatter, pCO$_2$, DMSsw). For each bloom, a nominal centre was identified, based upon maximum DMSsw and Chl-$a$ concentrations, and marked by deployment of a Spar Buoy. Repeat activities at the bloom centre included characterisation of the surface mixed layer by vertical profiling, collection of SSM samples at distance from the main vessel, and gradient flux on a catamaran. Overnight mapping was carried out to determine changes in bloom magnitude and position. The SOAP voyage was nominally divided into three different bloom periods (see Figure 4), with an initial dinoflagellate bloom (B1) located 12 hours into the SOAP voyage that exhibited strong biogeochemical signals, a coccolithophore bloom (B2) with initially moderate signals that weakened, and a final bloom (B3) of mixed community composition. Following a storm the surface water column structure and biogeochemistry were significantly different, and so this bloom was subdivided into B3a and B3b.

### 3.2.1 Environmental conditions during the SOAP voyage

Back-trajectory analysis of particle density was calculated for each bloom using the Lagrangian Numerical Atmospheric-dispersion Modelling Environment (NAME) for the lower atmosphere (see Figure 5). The meteorological situation evolved over the SOAP voyage from a high-pressure system with light winds during B1, to stronger winds during B2 and B3. The main weather features included a depression crossing the central South Island on DoY 54-55 during B2, and a second depression from the east from DoY 58 onwards. During B3 a vigorous front advanced up the east coast of the South Island



on DoY 61 with strong SW winds of 20 m s$^{-1}$, followed by a depression crossing the lower North Island on DoY 63 that maintained a fresh southerly airflow for the remainder of the voyage. Air and water temperatures during B1were generally similar indicating near-neutral stability, whereas B2 experienced

a period of warm, moist air and reversal in direction of turbulent heat fluxes, followed by a short period when air temperatures were 2-3$^{\circ}$C higher on DoY 56-58 (See Figure 6). Waves were dominated by swell from the south-southwest, with significant wave height mirroring trends in wind-speed, reaching a 5m maximum during the localised storm on DoY 61 (see Figure 6). Wave parameters obtained from NOAA WaveWatch III analyses indicated that wave height was 23% lower during B1 and B3, and 13% lower at

B2, relative to wave height south of New Zealand at 50$^{\circ}$S.

Table 1 summarises the hydrographic and biogeochemical characteristics in the surface mixed layer of the three phytoplankton bloom regions. B1 was a large dinoflagellate bloom with high surface DMSsw (maximum ~30 nmol L$^{-1}$; mean 16.8 nmol L$^{-1}$; Bell et al., 2015), and Chl-$a$ (maximum 3.4 mg m$^{-3}$), and significant CO$_2$ undersaturation with a mean surface pCO$_2$ of 320 ppmv (see Table 1). B1 was located,

south of the Mernoo bank, a deep channel between the western end of Chatham Rise and the east coast of the South Island. This region has been previously identified as a prime location for phytoplankton blooms, due to eddy-driven mixing and flow reversals arising from current and topographic interaction, which enhance iron and nutrient supply (Boyd et al., 2004). During B1, winds remained light (see Table 1) with a calm sea state, and the spar buoy drifted north-east primarily under the action of surface

currents. Solar irradiance was high and a shallow surface mixed layer developed (see Figure 6), with a significant near-surface temperature gradient (Walker et al., 2016). Mean nitrate and phosphate concentrations (5.3 and 0.4 µmol L$^{-1}$, respectively) were sufficient for phytoplankton growth, whereas silicate was low (see Table 1), and close to growth–limiting concentrations (Boyd et al., 1999). Although dinoflagellates dominated, coccolithophores biomass was higher at some stations, and nanoeukaryote

abundance was generally low. B1 was occupied for 5-6 days, during which broader regional excursions with overnight mapping identified a bloom of high Chl-$a$ but relatively low DMSsw to the south-west.

The vessel re-located to a coccolithophore bloom, B2, evident at the eastern end of the Chatham Rise in MODIS true colour satellite images (see Figure 4b). Upon arrival on DoY 52 B2 showed initial mean DMSsw of 9 nmol L$^{-1}$) and elevated Chl-$a$, and was characterised by a relatively warmer, shallower, saltier

surface mixed layer of lower nitrate concentration (compared to B1, see Table 1), typical of subtropical water. This appeared to provide optimal conditions for coccolithophores as surface water backscatter ($\beta_{660}$) was initially elevated by high lith abundances, with coccolithophores accounting for up to 40% of phytoplankton carbon. However, intrusion of warm, moist air associated with north-westerly winds, coincided with a reversal in the direction of turbulent heat fluxes, and was followed by a southwest wind



shift strengthening to 17 m s$^{-1}$ by DoY 56 (see Figure 6). This resulted in deepening and cooling of the
surface mixed layer with a corresponding increase in nutrient concentrations which, combined with a
decrease in solar irradiance, resulted in a decline in Chl-$a$ and DMSsw (Bell et al., 2015).

Following the 5-day occupation of B2, the vessel returned to the south of Mernoo bank to assess a
bloom that had developed near the original site of B1. Surface biogeochemical signals were initially weak

in B3a, with a mixed community of coccolithophores and dinoflagellates and low DMSsw (2.2 nmol L$^{-1}$)
and Chl-$a$ (mean 0.39 mg m$^{-3}$). However, an intense front advanced up the South Island and resulted in
strong SW winds that exceeded 20 ms$^{-1}$ (see Figure 6), after which mixed layer depth and associated
nutrients increased. Consequently, stations before and after the storm were physically and
biogeochemically disparate. B3a stations exhibited similar sea surface temperature to B1, but with a

deeper surface mixed layer and a Chl-$a$ half that of B1, whereas B3b stations were significantly cooler
(at 13$^{o}$C) and deeper (41m) than B1 (see Figure 7), with higher silicate concentration due the enhanced
vertical mixing. Subsequent stabilisation of the surface mixed layer by light winds combined with
elevated nutrients stimulated Chl-$a$, diatom and coccolithophore abundance in the final B3b stations
(see Figures 6 and 7).

## 4.    SOAP work programmes and observations

Measurements of parameters were carried out in three interlinked work programmes during the SOAP
voyage, as indicated in Figure 8 and detailed below.

### 4.1. The distribution and composition of aerosols, precursors and trace gases in the MBL

Aerosol concentration, size distribution, composition, water uptake and CCN concentration were
measured semi-continuously during SOAP to address the overall paucity of aerosol observations, and
the apparent rarity of nucleation events, over the remote ocean. These were characterised by a suite of
instruments covering a particle size range of 0.01 to 10 µm (see Figure 9 and Suppl. Table 1), which
enabled determination of the size-dependent contribution of PMA and nssSO4 to aerosol and CCN

concentrations. Aerosol characterisation identified variable Aiken and consistent sub-micron sized
accumulation and coarse modes, with the sub-micron aerosol mass dominated by secondary aerosol
with ammonium sulfate/bisulfate under light winds, and with an increase in sea-salt proportion as local
winds increased. Ongoing data analysis is examining whether significant nucleation events occurred.

The operational mode for underway aerosol measurement was to slowly steam at 1–2 kts into the

prevailing wind, across an area of high biological productivity or significant air-sea gas gradient,
generally between noon and 2:00PM when solar irradiance was maximal. The common aerosol inlet





developed during PreSOAP allowed uncontaminated air from above the bridge to be sampled when the wind was on the bow, so minimizing interference from ship stack emissions. Contamination events were screened out using a real-time clean-sector sampling "baseline" flag and switch (Harvey et al., to be

submitted), enabling clean collection of integrated samples. Although the vessel exhaust was the primary contaminant, other potential sources included the workboat and recirculation of polluted air around the ship, and longer range terrestrial influences were also assessed. Measurements of black carbon using an aethalometer, and $CO_2$ by high precision Cavity Ring-Down Laser Spectroscopy (CRDS) provided two independent variables for detecting contamination events, and some VOCs, measured by

PTR-MS (see Suppl. Table 1), were also used as indicators of diesel combustion. The vessel was orientated into the wind as often as possible, which resulted in a high frequency (~75%) of baseline sector conditions during the SOAP voyage. Clean marine air periods were defined post-voyage using the baseline wind sector (225 – 135° relative to bow and wind speed greater than 3 m s$^{-1}$), black carbon concentrations (less than 50 ngm$^{-3}$), and back trajectories indicating minimal terrestrial impact (periods

when the minimum number of hours over land in 72-hour back trajectory is zero), with periods of workboat operations also removed. An ensemble of Hybrid Single-Particle Lagrangian Integrated Trajectory (HYSPLIT) model back trajectories (Draxler and Rolph, 2013) was run for each hour of the voyage, and NAME back trajectories calculated for every three hours (Figure 5, Jones et al., 2007). Figure 10 shows particle number and CCN concentrations, compared to the number of hours the 72-hour back

trajectory spent over land calculated from HYSPLIT trajectories. Particle concentrations were generally higher during periods of terrestrial influence (see DoY 52 and 60, Figure 10), with average particle number concentrations of 1122 ± 1482 cm$^{-3}$, double that observed for clean marine air. Ion beam analysis also revealed the presence of silicate and aluminium on ambient submicron filter samples suggesting a terrestrial source, supporting the back-trajectory modelling of continental outflow.

During the initial occupation of B1 under light winds, the particulate matter (PM10) total ion mass was 9.5 μg m$^{-3}$ compared to subsequent samples under higher winds in the range 20-50 μg m$^{-3}$. The dominant components of the inorganic mass fraction were sea-salt ions and nssSO4, although a measurable organic fraction was also present (see below). The NaCl mass in light winds during B1 was 6.6 μg m$^{-3}$ with >95% of > 3μm diameter, relative to 32.5 μg m$^{-3}$ under stronger winds during B3b.

Although 72% was > 3μm, the largest difference in mass occurred in the 1.5 to 3μm size range. In contrast, the mass of nssSO4 was predominantly sub-micron sized; B1 exhibited the largest nssSO4 mass at 2.0 μg m$^{-3}$ with 85% in sizes <1μm, whereas in B3b, the nssSO4 mass was much lower at 0.6 μg m$^{-3}$ with 76% in <1μm sizes. These results confirm the influence of both physical and biogeochemical processes on aerosol composition.



Voyage particle number concentrations during clean marine periods averaged 534 ± 338 cm$^{-3}$, with CCN

concentrations of 178 ± 87 cm$^{-3}$ (±1 sd), and an average particle fraction activated into CCN of 0.4 ± 0.2.

Bloom average particle number concentrations ranged from a minimum of 385 ± 96 cm$^{-3}$ in B3b to a

maximum 830 ± 255 cm$^{-3}$ at the start of B2 (Figure 10). B1 displayed the highest CCN activation ratio, of

0.5 ± 0.2, potentially the combination of low wind speeds, large biogeochemical signals and SMA fluxes.

This suggests that particle growth from secondary organics may have enhanced the proportion of CCN

over B1. In contrast, the average CCN activation ratio for B3a was 0.13 ± 0.06. Nucleation mode particles

(10 nm and 15 nm), were measured by ultra-fine organic tandem differential mobility analyser (UFO-

TDMA, Vaattovaara et al., 2005), and Aitken mode particles (50 nm), by UFO-TDMA and a volatility and

hygroscopicity tandem differential mobility analyser (VH-TDMA, Johnson et al. 2004a; Villani et al.,

2008). This analysis typically identified a significant (up to 50% volume fraction) secondary organic

component during sunny conditions in bloom regions, particularly during B1. The TDMA results provided

further evidence for secondary organic aerosol processing of the dominant secondary nssSO4 mode

during B1. Deliquescence measurements (VH-TDMA) indicate that the Aitken mode population is largely

comprised of neutralised nssSO4 i.e. ammonium sulfate. Small and sporadic contributions to the Aitken

mode from a non-hygroscopic (number fraction up to 0.4) and a highly hygroscopic component (number

fraction up to 0.3) were observed in addition to the secondary nssSO4 mode (number fraction of 0.6 -

1). The water uptake and volatility of the sporadic highly hygroscopic mode indicates this may be

composed of PMA.

The *in-situ* aerosol size, number and composition measurements in the MBL were complemented by *in*

*vitro* chamber measurements of nascent SSA, to determine the PMA organic volume fraction and water

uptake properties. Nascent SSA filter samples were analysed using Fourier Transform InfraRed

spectroscopy (FTIR) for organic functional groups (Russell et al. 2011), and ion beam analysis for

inorganic concentrations (Cohen et al. 2004). Measurements of the hygroscopic growth factor and the

volatile fraction up to 450°C for 50-150 nm particles using the VH-TDMA were compared with those of

reference inorganic samples (e.g. sea salt, ammonium sulfate) to determine their organic volume

fractions (Modini et al. 2010). Complementing the VH-TDMA, the UFO-TDMA provided further

information on the organic content of particles of 50nm and down to 10 nm. The bubble chamber

observations indicate that the PMA contained a substantial primary organic fraction. VH-TDMA results

indicate that the Aitken mode PMA was primarily non-volatile (78-93%), with an average organic volume

fraction of 51% (ranging from 39 to 68%), and the UFO-TDMA results show an OVF ranging from 35-45%.

These results are consistent with observations in the North Pacific and Atlantic, for which an Aitken

mode volatile fraction of the order of 15% and OVF of 0.4-0.8 have been observed (Quinn et al. 2014).



FTIR analysis indicated that the POA aerosol in the chamber experiments was largely composed of hydroxyl functional groups, with minor contributions from alkanes, amines and carboxylic acid groups, consistent with previous PMO observations (Russell et al. 2011).

Although DMS was a primary focus of measurements during SOAP, a wide variety of other VOCs that potentially contribute to secondary organic aerosol formation were also measured. Halogens and halogen oxides were measured using Multi Axis Differential Optical Absorption Spectroscopy (Max-DOAS) and Electron Capture Detector-Gas Chromatography (ECD-GC). Iodine has been identified as a potentially important precursor of nucleation in coastal regions (Sellegri et al., 2005), and SOAP provided an opportunity to relate the presence of halogen oxides to phytoplankton biomass and composition in the surface ocean, and nucleation events in the MBL. A High Sensitivity Photon Transfer Reaction Mass Spectrometer (PTR-MS) measured continuously in H3O+ mode in the range of m/z 21- m/z 155 throughout the voyage (Lawson et al., to be submitted). Aldehydes, ketones and dicarbonyls were measured with 2,4-dinitrophenylhydrazine (2,4-DNPH) cartridges and high performance liquid chromatography (HPLC; Lawson et al., 2015), and a range of VOCs were sampled using adsorbent tubes and later analysed via Thermal Desorption-Gas Chromatography - Flame Ionisation Detection - Mass Spectrometry Detection (TD-GC-FID/MS). These measurements identified a positive relationship between DMS (m/z 63), acetone (m/z 59) and methanethiol (m/z 49), indicating common biological drivers (Lawson et al., to be submitted).

The first in situ measurements of aqueous phase SMA precursors dicarbonyls, glyoxal and methylglyoxal were obtained over the remote Southern Ocean during SOAP (Lawson et al., 2015). Parallel measurements of known dicarbonyl precursors, measured by PTR-MS, were used to calculate the expected yields of glyoxal and methyl glyoxal, which accounted for < 30% of observed mixing ratios indicating an unidentified source of dicarbonyls (Lawson et al., 2015). This was corroborated by inclusion of SOAP glyoxal measurements obtained by Max-DOAS measurement in a global database, which concluded that the missing glyoxal source was an order of magnitude greater than identified sources (Mahajan et al, 2014). Surface mixing ratios of glyoxal converted to vertical columns, were significantly lower than average vertical column densities (VCDs) from satellite retrievals, possibly reflecting the difficulty of retrieving low glyoxal VCDs over the ocean, or incorrect assumptions about the vertical distribution of glyoxal in the atmosphere (Lawson et al., 2015).

## 4.2. Rates and controls of volatile and precursor emissions at the air-sea interface

DMS measurements were made using three different instruments (see Suppl. Table 1); an Atmospheric Pressure Ionisation-Chemical Ionisation Mass Spectrometer (API-CIMS) continuously monitored DMS in



both phases (Bell et al., 2015), a PTR-MS monitored DMSa (Lawson et al., to be submitted), and discrete water measurements were made by a Sulfur Chemiluminescence Detector Gas Chromatograph (SCD-GC; Walker et al., 2016). Intercomparison of sulfur measurements is not easily or routinely performed (Bell et al., 2012), but it was important to intercalibrate the three DMS instruments to ensure robust data comparability (Walker et al., 2016). The SCD technique compared well with traditional gas

chromatography (and flame photometric detector) in a subsequent international intercalibration exercise for dissolved DMS (Swan et al., 2014).

Although the majority of DMS flux estimates to date have been derived by applying an independently determined transfer velocity ($k$) to the measured DMS gradient at the ocean surface ($\Delta$DMS), there has been a recent increase in direct micrometeorological measurements of DMS flux. Measurements at 10-

30-minute resolution show considerable variability in flux, which may reflect methodological artefacts or inherent variability in the distribution of DMS. SOAP provided a platform for comparing eddy covariance (EC) flux measurements of DMS using API-CIMS (Bell et al., 2015), with a gradient flux technique using a drogued catamaran within one kilometre of the vessel (Smith et al., to be submitted). The gradient flux technique is less direct than EC but provides an alternative reference on a platform

that is relatively free of shipboard air-flow distortion. The EC system sampled from an intake on the ships bow, with flux instruments mounted on the foremast 12.6m above sea level, and the air pumped to a containerised laboratory on the foredeck. Additional meteorological measurements were obtained from a weather station above the bridge. Both sites are subject to airflow distortion which is azimuthally dependent (Popinet et al., 2004). The catamaran sampling framework, which consisted of four air

intakes distributed vertically on a 5.3m mast, sampled closer to the water surface where gas gradients are largest. Flux measurements were augmented by continuous near-surface measurement of physical parameters using a range of sensors attached to a Spar Buoy, with stratification determined by temperature sensors at 0.5 m intervals (Walker et al., 2016), turbulence determined by a Vector acoustic Doppler Velocimeter at 0.6 m depth. This permitted comparison of $k_{DMS}$ estimates with near-surface

upper-ocean turbulence at a distance from the vessel (Smith et al., to be submitted). Wave-breaking whitecap coverage was monitored using a Campbell Scientific 5-megapixel camera (cc5mpx) located on the starboard side of the vessel (Scanlon and Ward, 2016). This provided an indicator of bubble entrainment, which contributes to the differential transfer rate of DMS and $CO_2$ due to their different solubilities (Blomquist et al., 2006; Bell et al., submitted).

Although SOAP primarily focussed on DMS fluxes, EC measurements of $CO_2$ flux were an important adjunct measurement for providing insight into gas exchange mechanisms and controls, and improving gas transfer algorithms for gases of differing solubilities. Both open and closed path eddy covariance





flux measurements were made for $CO_2$ during SOAP, following the initial trials on PreSOAP; this enabled ship motion and airflow distortion to be examined in addition to technical artefacts (Landwehr et al., to

be submitted). Comparison of EC measurements with wet and dry incoming gas streams, and an empirically-based post-processing correction, indicated that only gas stream drying produced robust $CO_2$ flux and $k_{CO2}$ estimates (Landwehr et al., 2014). Measurement of DMS and $CO_2$ fluxes also provided further constraint of k parameterisations based upon wind-speed, and the opportunity to assess the influence of bubbles on gas exchange at high wind speeds (Bell et al., submitted). DMS fluxes derived by

EC and gradient flux techniques showed good agreement (Bell et al., 2015; Smith et al., to be submitted), and confirmed previous observations that gas transfer is a linear function of wind-speed at low to intermediate winds (Blomquist et al., 2006; Yang et al., 2011). However, despite winds reaching 20 m s$^{-1}$ during the latter part of SOAP insufficient data was obtained to draw conclusions regarding the reported deviation of $k_{DMS}$ under high winds (Bell et al., 2015). However, SOAP provided a novel estimate

of the size of the EC flux footprint, and the temporal-spatial mismatch between DMSsw and shipboard measured fluxes, highlighting the importance of considering skew in flux estimates arising from non-linear distribution of DMSsw (Bell et al., 2015).

A further objective of SOAP was comparison of measured DMS fluxes with calculated estimates from the COAREG model (Fairall et al. 2011) based on ΔDMS, to assess potential discrepancies with modelled

fluxes (Marandino et al., 2008; Walker et al., 2016). Potential factors examined here included air and water stability, and the influence of the SSM. Despite the agreement between DMS flux estimates by the two micrometeorological techniques, there was significant departure from COAREG predictions (Fairall et al., 2011) on occasions, suggesting the influence of unidentified processes (Smith et al., to be submitted). One potential example was the suppressed DMS flux during a period of atmospheric stability

and reversed heat flux during B2. Concurrent EC flux measurement for DMS and $CO_2$ also provided an opportunity to assess other influences on $k$. The DMS flux data indicate that the $k_{DMS}$–wind speed relationship was relatively insensitive to surface biogeochemistry or wave action during SOAP (Bell et al., 2015). In addition, data from SOAP were used to parameterise whitecap coverage against wind-speed, and assess the influence of maturing waves on the quantification of breaking waves (Scanlon and

Ward, 2016).

**4.3. Surface ocean biogeochemical influences on aerosols and volatiles**

Surface mapping of DMSsw and pCO2, using API-CIMS and IRGA, respectively (Bell et al., 2015) were critical to the SOAP voyage strategy and the aims of the two workpackages discussed above. These measurements also provided insight into the covariance of DMS sources and $CO_2$ sinks in surface waters,

and established the importance of this region to global budgets. The New Zealand Coastal province



(NEWZ), which includes the frontal region (STF) studied during SOAP, is characterised in the global DMS climatology by year-round low DMS concentrations with a maximum <2 nmol L$^{-1}$ (Lana et al., 2011). This infers that this region has some of the lowest global DMSsw concentrations, in marked contrast to the adjacent South Subtropical Convergence (SSTC) province, which occupies the remainder of the 35-50°S latitude band and accommodates the STF, which is characterised by a mid-summer maximum of 10 nmol L$^{-1}$ DMS. This discrepancy between the two regions likely reflects the low number of DMS observations for the NEWZ province in the climatology (n=6; Lana et al., 2011). Previous DMSsw measurements in subantarctic waters south of the Chatham Rise, and east of Tasmania in the SSTC biome (Archer et al., 2011; Griffiths et al., 1999), are consistent with this climatological estimate, whereas larger unpublished surveys have recorded elevated surface DMSsw during austral spring (October 2000), with a mean DMSsw of 4.5 (+/- 6.8) nmol L$^{-1}$ on the Chatham Rise (Harvey et al., pers. comm). Combining these measurements with data from the SOAP campaign (mean DMSsw = 6.6 nmol L$^{-1}$) gives a weighted-mean DMSsw of 5.3 nmol L$^{-1}$ (n=5300, see Table 2), confirming that DMSsw in the NEWZ province is currently underestimated, and is in fact more typical of the SSTC province. This is supported by the EC flux measurements during SOAP, which recorded maximum and mean fluxes of 100 and 16.3 µmol S m$^{-2}$d$^{-1}$, respectively, (Bell et al., 2015), which exceed the climatological mean of >10 µmol S m$^2$ d$^{-1}$ for the SSTC region (Lana et al., 2011). In addition, the high MBL DMS concentrations of 1000 ppt recorded during SOAP exceed DMSa at coastal stations on the New Zealand North Island in summer (Harvey et al., 1993; de Bruyn et al., 2002; Wylie and de Mora, 1996). Although seasonally constrained, the SOAP measurements provide evidence that regional DMS emissions are significant in the South West Pacific. The large dataset of regional concentrations and flux will allow further refinement of global climatologies, such as the Global Surface Water DMS Database and the Surface Ocean CO2 Atlas (SOCAT).

The spatial variability of DMSsw was related to surface ocean biogeochemistry and bloom type by measurement of a suite of ancillary parameters in underway mode, including temperature and salinity, Chl-*a,* chromophoric dissolved organic matter *(*CDOM), β$_{660}$ backscatter, dissolved oxygen and pCO$_2$ (see Suppl. Table 1). The vertical and lateral variability of DMSsw, and the dissolved and particulate pools of its precursor DMSP, were quantified in the surface mixed layer, and related to plankton biomass and community composition, nutrient and organic composition and physical drivers including irradiance (see Suppl. Table 1). Process studies of DMSP cycling included deckboard incubations examining the bacterially-mediated pathways of DMSP cleavage and demethylation in relation to different bloom dynamics (Lizotte et al., submitted). DMSP concentrations were relatively high, reaching a maximum of 160 nmol L$^{-1}$, and showed significant correlation with phytoplankton biomass during SOAP. However,



the yield of DMS from bacterial conversion of dissolved DMSP was variable with no spatial trend,
although a correlation with leucine incorporation indicates that DMSP was an important carbon source
for bacteria. Overall, gross DMS production by bacteria in deckboard incubations of near-surface water
was relatively low, at < 6 nmol $L^{-1}$ $d^{-1}$, inferring that phytoplankton-mediated conversion of DMSP was
likely a significant near-surface source of DMS (Lizotte et al., submitted).

The SSM is a potentially important interface controlling MBL and aerosol composition, as it is the
interface across which material exchanges between atmosphere and ocean. Physical and
biogeochemical processes within this thin layer have the potential to alter transfer via factors, such as
the concentration of organic material and enhanced biological and photochemical processing. Near-
surface $CO_2$ gradients have been observed (Calleja et al., 2005), and several studies report DMS
enrichment in the SSM (see summary in Walker et al., 2016). If DMS consumption or production in the
SSM is significant this represents a potential source of discrepancy in comparison of measured fluxes
with that calculated by the COAREG model (see above). The biogeochemistry of the SSM and the upper
1.6m surface water were characterised at 10 stations during SOAP at distance from the research vessel,
to determine the spatial variability in composition within, and between, different phytoplankton blooms
(Walker et al., 2016). Near-surface DMS gradients were generally negligible, except during B1 where low
wind-speed, near-surface stratification and high dinoflagellate abundance may have combined to
enhance DMS in the SSM relative to subsurface waters. The observed DMS enrichment factors in the
SSM during B1, ranging from 1.4 to 5.3, are some of the highest reported to date. The anomaly between
measured DMS fluxes and COAREG estimated was also greatest during B1, inferring that DMS emissions,
and associated k-wind-speed parameterisations, may be sensitive to DMS production in the SSM under
certain conditions. However, the observations also raise questions as to how such significant DMS
enrichment is maintained in the SSM, as high DMS production would be required to ameliorate loss
processes (Walker et al., 2016).

**5. Conclusions**

The SOAP voyage has identified new questions in important areas of SOLAS-related research, including
the influence of the SSM, implications for secondary aerosol formation, and unidentified sources of
organic aerosol precursors, all of which are potentially influenced by photochemistry in the surface
ocean and MBL (Lawson et al., 2015). It has also provided insight into the relative importance of PMA
and SMA formation in the MBL and entrainment from the free troposphere (Raes, 1995), and addressed
confounding technical challenges including small-scale heterogeneity in surface waters, clean air



baseline sampling, and discrepancies between existing techniques and models. An overarching aim of the SOAP campaign was to assess potential relationships between surface water biogeochemistry and corresponding or related species in the MBL, to identify the factors influencing aerosol precursors, and their potential as analogues. Chl-*a* is a general indicator of biological productivity and is readily

retrievable by satellite, and consequently has been investigated as a potential proxy for DMSsw (Lana et al., 2011). The SOAP voyage provided a platform to validate this observation, particularly as it took place in the 40-60°S latitude band which exhibits the most significant regional correlation between Chl-*a* and DMSsw (Vallina et al., 2006). Correlations were apparent during SOAP between Chl-*a* and DMSP (Lizotte et al., submitted), and Chl-*a* with DMSsw and DMSa, but there was no corresponding

relationship between Chl-*a* and DMS flux (Bell et al., 2015). Correlations have been reported previously for Chl-*a* with CCN (Meskhidze and Nenes, 2006), and aerosol organic enrichment (Gantt et al., 2011), although other assessments have shown variable results (Russell et al., 2010; Rinaldi et al., 2013). The measurement of PMA and SMA composition and number coincident with multi-species characterisation of MBL and surface water composition during SOAP has provided a broad database with which to assess

and develop these relationships for potential application in remote sensing and Earth System Models. The first step towards this is the inclusion of SOAP aerosol and tropospheric data in the global ACCESS-UKCA model (Woodhouse et al., 2015), containing the GLOMAP-mode aerosol scheme (Mann et al., 2010, 2012), which shows very good agreement with observed distributions of condensation nuclei (Woodhouse et al, pers. comm.)

### 6. Data availability

The underway DMSsw can be downloaded at http://saga.pmel.noaa.gov/dms/select.php.

### 7. Supplement link (to be provided by Copernicus)

### 8. Author contribution

The SOAP campaign was led and coordinated by CSL, MJS & MJH. All authors developed analytical methods and instruments used during the SOAP campaign, and also analysed and interpreted data. CSL, MJS, MJH, TGB, LTC, FCE, SJL, ML, AM, JM, KAS, PV and CFW made measurements during the SOAP
voyage. CSL led manuscript preparation, with contributions from MJS, MJH, TGB, LTC, SJL, ML, RZ, PV and CFW.

The authors declare that they have no conflict of interest



## 9.    Acknowledgements

We acknowledge the invaluable assistance of the Captain, officers and crew of the R/V *Tangaroa*. We thank Kim Currie, Steve George, Matt Walkington, Mark Gall, Marc Mallet, Greg Olsen, Gus Olivares and Nick Talbot for collection and analysis of samples; Christa Marandino, Warren DeBruyn and Cyril McCormick for assistance with API-CIMS measurements; Sebastian Landwehr, Scott Miller and Brian Ward for $CO_2$ flux measurements; Paul Johnson, Karin Kreher and Neil Harris for halocarbon and VOC measurements; Jason Ward for assistance with CCN measurement; Gordon Brailsford for Picarro set up; Paul Selleck and Min Cheng for analysis of VOC samples, Sarah Connors for the back-trajectory suite using the UK Met Office NAME model, and Matt Woodhouse for analysis of SOAP data using the global ACCESS-UKCA model. We also acknowledge Maurice Levasseur and Melita Keywood for their comments, and their support of the microbial DMSP cycling and VOC measurements, respectively.

## 10.    Funding

This research was supported by funding from NIWA's Climate and Atmosphere Research Programme 3 – Role of the oceans (2015/16 SCI), and a Postdoctoral Fellowship (CO1X0911) for CW from the New Zealand Ministry for Business, Innovation and Employment (MBIE). Flux measurements were supported by a NSF Atmospheric Chemistry Program (grant nos. 08568, 0851472, 0851407 and 1143709), and VOC and CCN measurements supported by CSIRO's Capability Development Fund. PV participation was supported by EU COST action 735, the Academy of Finland through the Centre of Excellence, and via a Finnish Academy visiting grant no. 136841. SL participation was supported by Science Foundation Ireland as part of the US–Ireland R&D Partnership Programme under grant number 08/US/I1455, and a SFI Short-term Travel Fellowship under grant 09/US/I1758-STTF-11. We thank the Natural Sciences and Engineering Research Council of Canada (NSERC) for supporting the microbial DMSP cycling research.

## 11.    Tables


Atmospheric Chemistry and Physics Discussions — Open Access

| Bloom | Time/Location | | | | Meteorological | | | | Hydrodynamic | | | Biogeochemical | | | | |
|---|---|---|---|---|---|---|---|---|---|---|---|---|---|---|---|---|
| | Start NZST (DoY UTC) | End NZST (DoY UTC) | Bloom Centre Lat. | Bloom Centre Long. | Atmos Press. mb | Irrad W m$^{-2}$ | U$_{10}$ Range m s$^{-1}$ | Hs m | MLD m * | SST °C # | Sal. # | Nitrate/ Phosphate/ Silicate µmol L$^{-1}$ * | Chl-$a$ mg/m$^3$ * | pCO$_2$ ppmv # | DMSsw nmolL$^{-1}$ # | Dominant Phytoplankton * |
| B1 | 14/02/12 2:00 (44.6) | 19/02/12 12:00 (50.0) | -44.34 -44.61 | 174.2 174.78 | 1019.1 ± 2.9 | 232 <1061 | 6.6 (5-7.6) | 2.0 | 14.5 ±1 | 14.5 ±0.4 | 34.48 | 5.3 ± 0.9 0.43 ± 0.2 0.35 ± 0.1 | 0.84 ±0.2 <3.4 | 320 ±24 | 16.8 ±1.5 | Dinoflagellate |
| B2 | 21/02/12 16:15 (52.2) | 26/02/12 12:00 (57.0) | -43.55 -43.71 | 180.16 180.32 | 1011.5 ± 5.3 | 196 <1079 | 10.4 (6.9-12.4) | 2.9 | 24.0 ±9 | 15.8 ±0.2 | 34.6 | 1.7 ± 1.0 0.27 ± 0.07 0.41 ± 0.33 | 0.67 ±0.3 <1.0 | 339 ±9 | 9.1 ±2.9 | Coccolithophore Dinoflagellate |
| B3a | 27/02/12 10:00 (57.9) | 1/03/12 04:00 (60.67) | -44.11 -44.61 | 174.47 174.88 | 1010.0 ± 8.2 | 242 <1212 | 10.3 (8.1-12.1) | 2.6 | 28.6 ±1.7 | 14.4 ±0.2 | 34.32 | 3.7 ± 1 0.34 ± 0.06 0.3 ± 0.16 | 0.44 ± 0.17 <0.92 | 333 ±14 | 5.37 ±1.5 | Mixed |
| B3b | 02/03/12 06:00 (61.7) | 5/03/12 17:00 (65.2) | -44.19 -44.78 | 174.3 174.93 | 1008.6 ± 9.4 | 182 <1016 | 12.6 (8.5-14.9) | 3.6 | 41.1 ±6 | 13.2 ±0.4 | 34.32 | 4.2 ± 1.1 0.39 ± 0.1 0.48 ± 0.05 | 0.59 ±0.2 <1.1 | 340 ±8 | 3.1 ±1.2 | Mixed |

**Table 1.** Summary of surface water characteristics during each bloom period. All values are Mean ±1 standard deviation, except where maximum value also shown by <. * indicates value derived from 2-10m depth on all stations during bloom occupation; # indicates continuous measurement in surface waters (nominal 6m depth). Abbreviations: Lat: Latitude; Long.: Longitude; Atmos. Press.; Atmospheric Pressure; Irrad.: Irradiance; U$_{10}$: Wind speed adjusted to 10m height (uncorrected for vessel flow distortion); Hs: Significant wave Height; MLD: Mixed Layer Depth; SST: Sea Surface Temperature; Sal>: Surface salinity; Chl-$a$: Chlorophyll-$a$;



| Voyage | Date | Latitude | Longitude | Mean DMS (nmol L$^{-1}$) | Std Dev | N | Method | Reference |
|--------|------|----------|-----------|--------------------------|---------|---|--------|-----------|
| BOX | October 2000 | 39.5-47°S | 170-179°E | 4.55 | 6.8 | 482 | FPD-GC | this paper |
| | November 2005 | 49-50°S | 175°E | 1.75 | - | 2 | FPD-GC | Kiene et al., 2007 |
| SAGE | April 2006 | 41-46.6°S | 172.5-78.5E | 1.06 | 0.9 | 6 | PFPD | Archer et al., 2011 |
| PreSOAP | February 2011 | 42.5-44°S | 174E-178°W | 2.2 | 2.0 | 736 | MIMS | this paper |
| SOAP | March-April 2012 | 41.7-46.5°S | 172E-179°W | 6.36 | 4.4 | 4132 | API-CIMS | Bell et al., 2015 |
| SOAP | March-April 2012 | 41.7-46.5°S | 172E-179°W | 11.5 | 9.2 | 22 | SCD | Walker et al., 2016 |
| S.W. Pacific | Weighted Mean | 39.5-50°S | 170-179°W | 5.6 | | 5380 | | |
| NEWZ | | 35-55S | 170E-170W | 0.05-2.0 | | 6 | Climatology | Lana et al., 2011 |
| SSTC | | 35-50S | 170W-170E | 0.05-10 | | | Climatology | Lana et al., 2011 |

**Table 2.** DMS data for the S.W. Pacific region east of New Zealand.



## 12.    Figures

**Figure 1.** The SOAP voyage track in the Chatham Rise region, overlain by Sea Surface Temperature ($^o$C), with the study region (box) indicated in the inset bathymetric map of New Zealand.

**Figure 2.** nssSO4 concentrations at New Zealand coastal atmospheric monitoring sites

**Figure 3.** Continuous measurement during PreSOAP of a) wind speed (m s$^{-1}$), b) atmospheric DMS (ppt), c) surface water DMS (nmol L$^{-1}$), and d) surface chlorophyll-*a* (mg m$^{-3}$; quenched data removed).

**Figure 4**. 8-day composite images (at 4 km resolution) during the SOAP voyage for a) 10-17 Feb 2012 (DoY 41-48), b) 18-25 Feb 2012 (DoY 49-56), and c) 26th Feb-4th March (DoY 57-64), showing bloom locations (red dots), and daily true colour images for d) 16 Feb 2012 (DoY 47), e) 18 Feb 2012 (DoY 49) and f) 3 March (DoY 65) (MODIS Aqua data courtesy of NASA).

**Figure 5**. a)-c) Synoptic meteorology summary for each bloom period during the SOAP voyage. Surface pressure and wind plots are derived from the NZ local area Unified Model NZLAM, and the bloom location indicated by a red dot. d)-f) Back-trajectory analyses for each bloom period during the SOAP voyage. This was calculated using the Lagrangian Numerical Atmospheric-dispersion Modelling Environment (NAME) for the lower atmosphere (0-100m) as time integrated particle density (g s m$^{-3}$). Each plot shows the back-trajectory of 8 "releases", i.e. one every three hours over 24 hours for the actual ship position.

**Figure 6**. Meteorological and hydrodynamic variables during the SOAP voyage, including a) Wind speed (WS, ms$^{-1}$); b) Direction (Dir., $^o$); Wind (blue) and wave (cyan); c) Temperature (Temp.,$^o$C); Air (black) and surface water (green); d) Irradiance (Irrad., Wm$^{-2}$) and e) Significant wave height (Hs, m). The Bloom occupation periods are indicated by the red horizontal bars in the upper panel.

**Figure 7**. Surface water properties (2-10m) during the SOAP voyage: Temperature (Temp, $^o$C), Mixed Layer Depth (ML Depth, m), Chlorophyll-a (Chl-*a*, mg m$^{-3}$), and nitrate concentration (μmol L$^{-1}$), with the occupation period for each bloom indicated by the vertical shaded bars.

**Fig. 8**. SOAP parameters and integrated work programmes





**Figure 9**. Aerosol characterisation during SOAP indicating size spectral (red) and total counts (black).

**Figure 10**. a) Marine boundary layer CN concentrations (top, CPC3772 in blue, CPC3010 in red), b) CCN concentrations (middle) and c) number of hours 72 hour back-trajectory was over land (bottom, 27-member ensemble average).

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


**Figure 1.** The SOAP voyage track in the Chatham Rise region, overlain by Sea Surface Temperature (°C), with the study region (box) indicated in the inset bathymetric map of New Zealand.



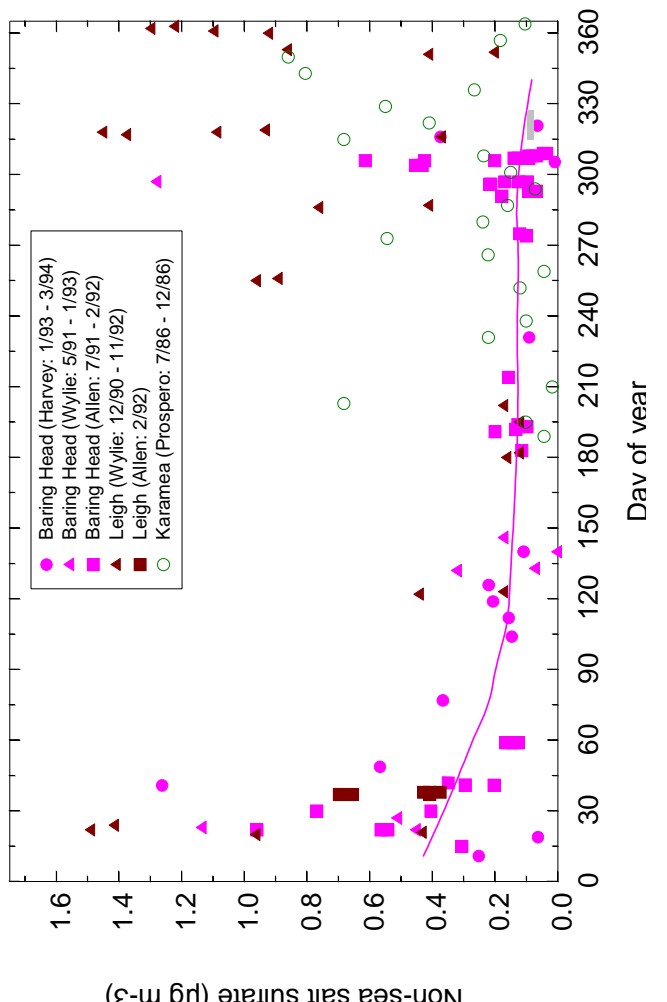

**Figure 2.** nssSO4 concentrations at New Zealand coastal atmospheric monitoring sites





**Figure 3.** Continuous measurement during PreSOAP of a) windspeed (m s$^{-1}$), b) atmospheric DMS (ppt), c) surface water DMS (nmol l$^{-1}$), and d) surface chlorophyll-$a$ (mg m$^{-3}$; quenched data removed).



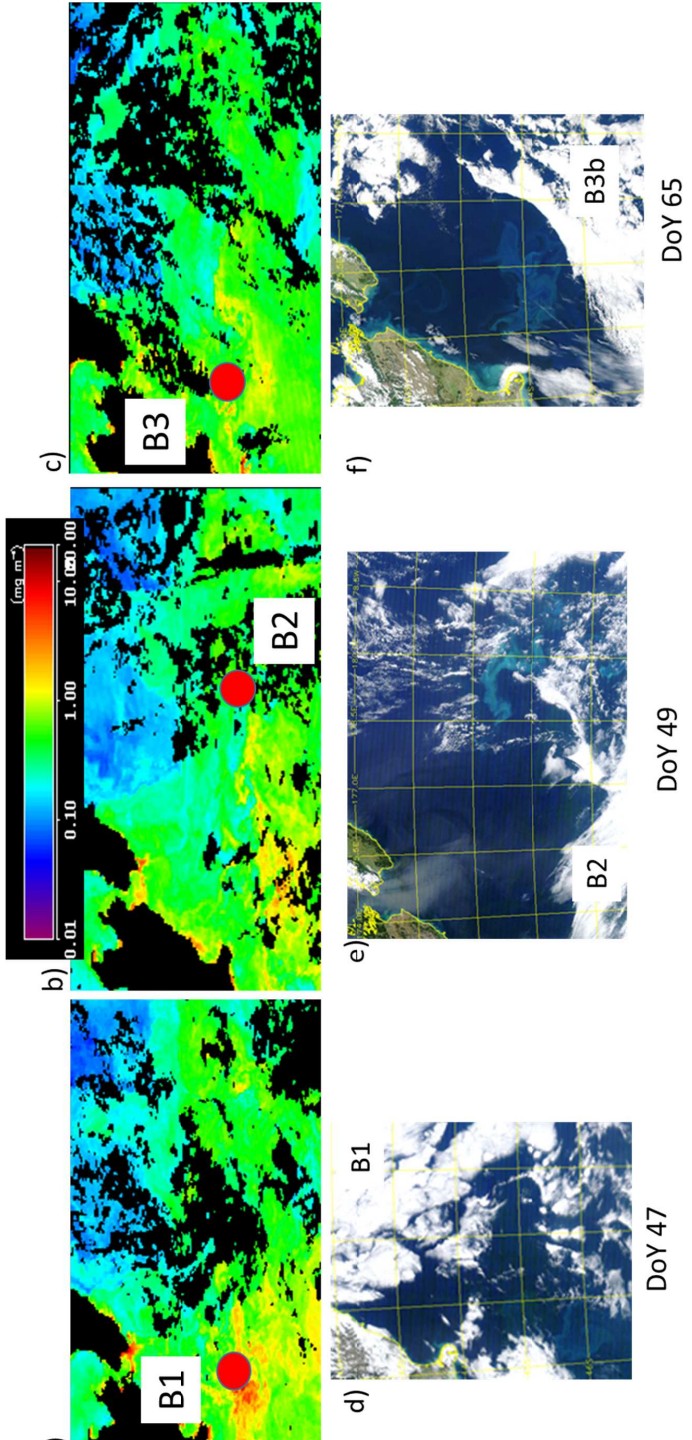

**Figure 4.** 8-day composite images (at 4 km resolution) during the SOAP voyage for a) 10-17 Feb 2012 (DoY 41-48), b) 18-25 Feb 2012 (DoY 49-56), and c) 26th Feb-4th March (DoY 57-64), showing bloom locations (red dots), and daily true colour images for d) 16 Feb 2012 (DoY 47), e) 18 Feb 2012 (DoY 49) and f) 3rd March (DoY 65) (MODIS Aqua data courtesy of NASA).



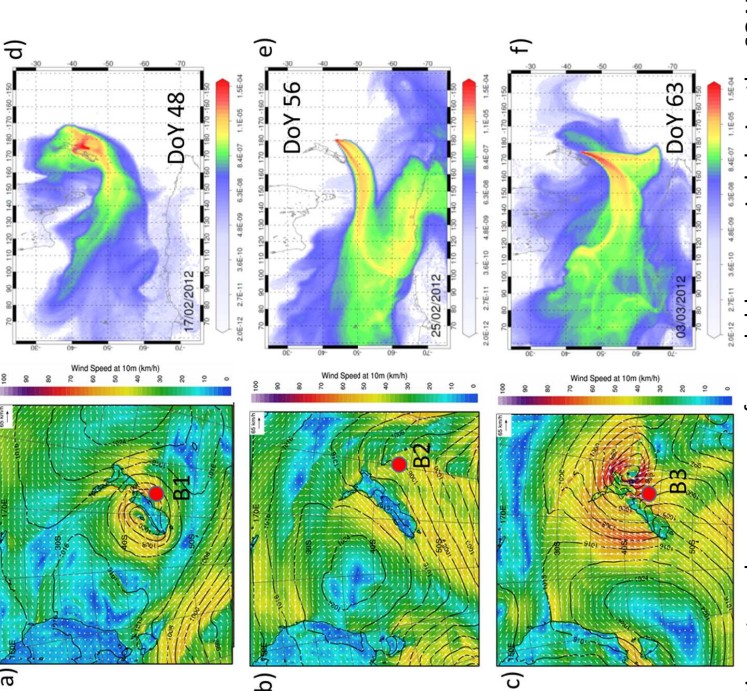

**Figure 5.** a)-c) Synoptic meteorology summary for each bloom period during the SOAP voyage. Surface pressure and wind plots are derived from the NZ local area Unified Model NZLAM, and the bloom location indicated by a red dot. d)-f) Back-trajectory analyses for each bloom period during the SOAP voyage. This was calculated using the Lagrangian Numerical Atmospheric-dispersion Modelling Environment (NAME) for the lower atmosphere (0-100m) as time integrated particle density (g s m$^{-3}$). Each plot shows the back-trajectory of 8 "releases", i.e. one every three hours over 24 hours for the actual ship position


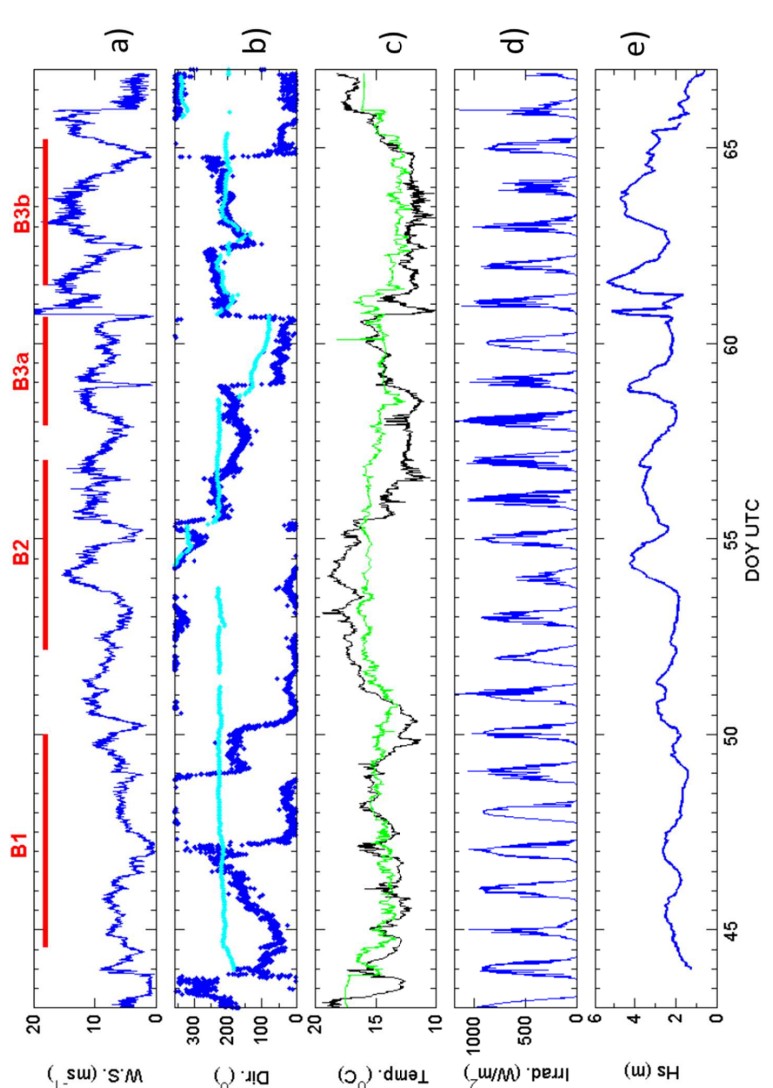

**Figure 6.** Meteorological & hydrodynamic variables during the SOAP voyage, including a) Windspeed (WS, m s$^{-1}$); b) Wind direction (Dir., °); c) Temperature (Temp., °C); Air (black) and surface water (green); d) Irradiance (Irrad., Wm$^{-2}$) and e) Significant wave height (Hs, m). The Bloom occupation periods are indicated by the red horizontal bars in the upper panel.

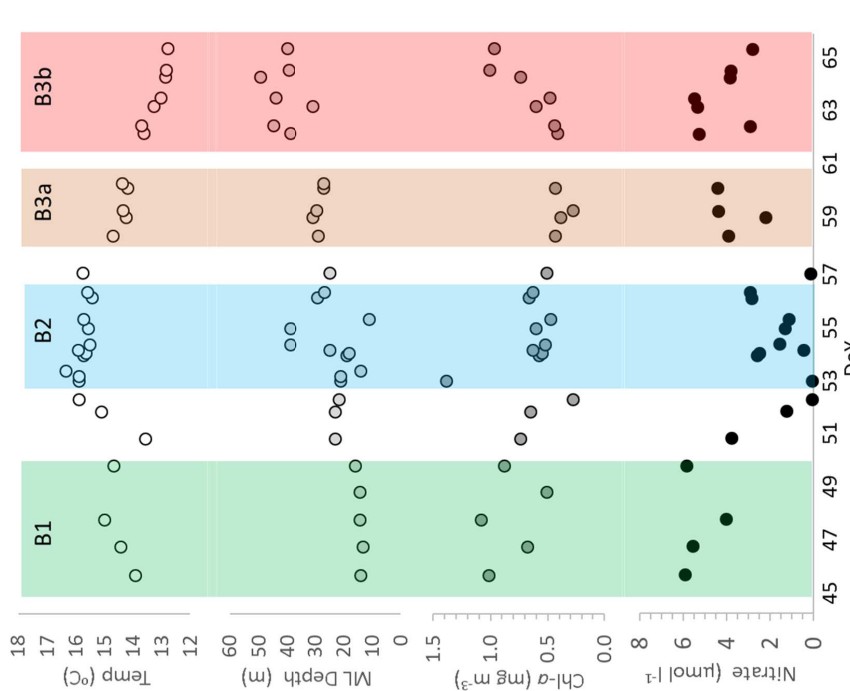

**Figure 7.** Surface water properties (2-10m) during the SOAP voyage: Temperature (Temp, °C), Mixed Layer Depth (ML Depth, m), Chlorophyll-a (Chl-a, mg m-3), and nitrate concentration (μmol l⁻¹), with the occupation period for each bloom indicated by the vertical shaded bars.




**Fig. 8.** SOAP parameters and integrated work programmes





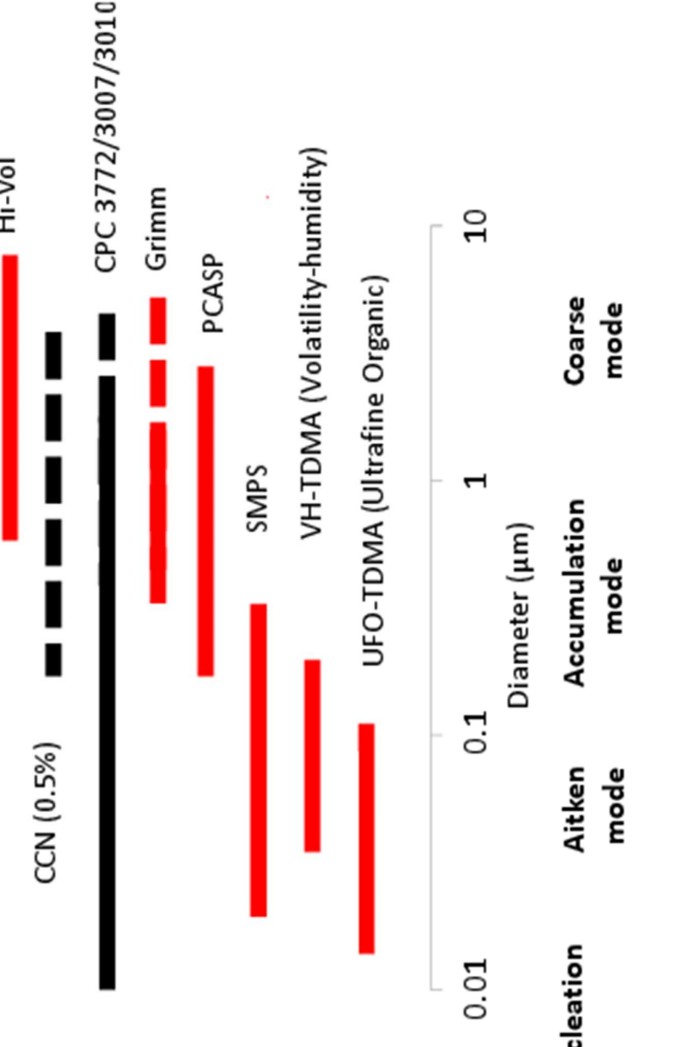

**Figure 9.** Aerosol characterisation during SOAP indicating size spectral (red) & total counts (black).





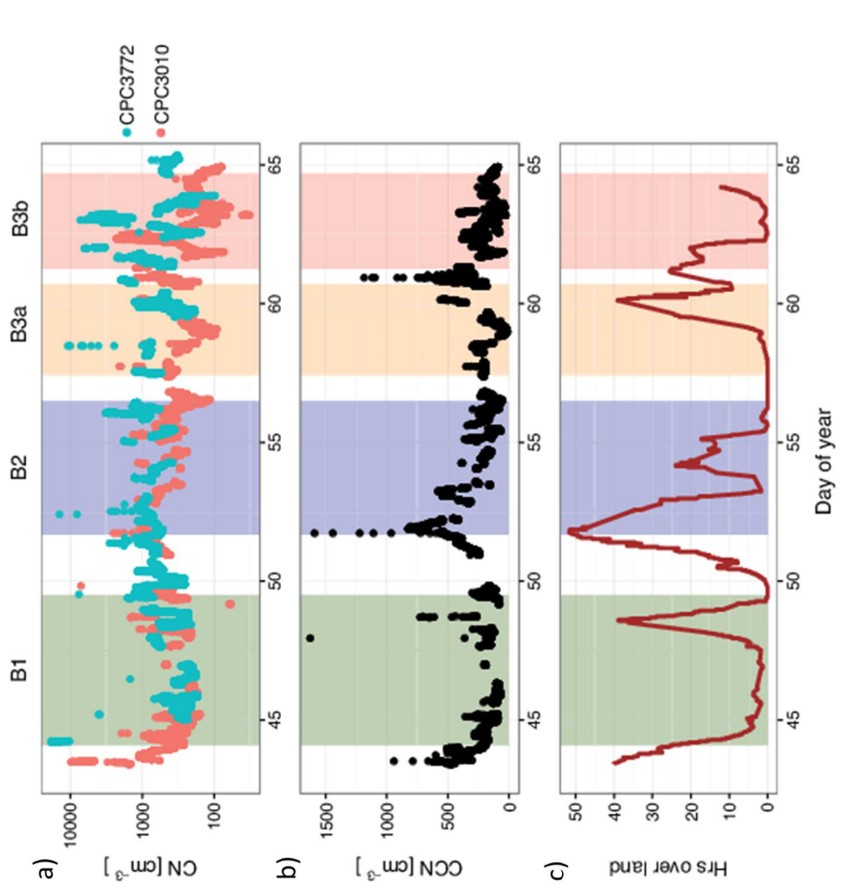

**Figure 10.** a) Marine boundary layer CN concentrations (top, CPC3772 in blue, CPC3010 in red), b) CCN concentrations (middle) and c) number of hours 72 hour back-trajectory was over land (bottom, 27-member ensemble average).