# Peer review of "Overview and Preliminary Results of the Surface Ocean Aerosol Production (SOAP) campaign"

_Atmospheric Chemistry and Physics, 2017_

## Referee Comment (RC1) · R. Simó (Referee) · 31 Jul 2017

The SOAP experiment is one of the largest, most comprehensive and most interesting efforts conducted so far to study biogeochemical surface ocean – lower atmosphere interactions. Some articles on topical studies within the global study have been or are being published, but there is the critical need for an overview paper like this that provides the context and describes the experimental approach. The present manuscript is definitely worth publishing to serve this purpose, even though it falls a bit short in enunciating the main findings and advances of knowledge.

I particularly like the introduction, which does a very good job with summarizing the state of the art, the gaps of knowledge and the need for such an experiment. The

oceanographic and environmental regional context is very much appreciated too. The other aspect I like best is the listing of the instruments and how they complement one another. This is something typically missing in many papers for a lack of space, and that the nature of this manuscript allows.

I do not have much to say. I miss comparison with previous similar cruises, such as the ACSOE or the SAGE, and statement of what is different and how SOAP goes a step forward.

In terms of a bloom-related study, SOAP is a bit disappointing. I mean, the links between each of the blooms, its biogeochemical processes, and the results of the air-sea exchange, are weak. Effort is made in the present manuscript to argument that each of the situations or blooms is not a static environment but dynamic, with changes associated with meteo forcing and so forth. This is sharp and honest – the drawback is that the blooms were not very clearly delineated so that process-based associations with aerosol precursors of more general applicability could be built. Do you the authors agree with this analysis? Along these same lines, the recent paper by Royer et al. (2016) in Scientific Reports shows dramatic changes in DMS concentration associated with the passage of a storm.

Specifics

-Line 207: Mahajan et al. 2006 should read 2015 -Line 314: remove parenthesis after 9 nmol L-1 -Page 17: when discussing about the underestimation of the current climatology for the region, and call for a revision into much higher concentration, to what extent do you think your numbers are biased high because you deliberately visited blooms? What can you say about average regional concentrations? -Page 19: To me, it is pretty obvious that instantaneous correlations between chla and the aqueous concentration of DMS or any other biogenic volatile can be expected (yet not always found), but not necessarily with the flux. The flux depends primarily on the aqueous concentration but also on e.g. the wind speed. Therefore, correlations between biological markers and

the emission flux are to be expected, if anything, over longer time scales.

---

## Referee Comment (RC2) · Anonymous Referee #2 · 1 Aug 2017

This manuscript provides an overview of the multi-disciplinary SOAP cruise off the coast of New Zealand in 2012. I believe such an overview is important and that the manuscript should be published with the following modifications:

1. Line 37 You don't show a correlation between chlorophyll-a and DMSsw.

2. Line 80 You are mixing aerosol mass and number here.

3. Line 173 What is secondary production?

4. Line 199 Should read "aerosols and their precursors".

5. Line 263 Could you please give more details on the bubble chamber.

6. Line 263. The Supplementary table should be in the main manuscript. It would be

helpful to have a reference for each measurement.

7. Line 276. What do you mean by "biogeochemical signals"?

8. Figure 5 needs to be larger to make it more readable.

9. What is the light blue line in Figure 6b?

10. Line 340. Aerosol number concentration. . ..

11. Line 386. CCN data should include the % supersaturation. Were all measurements made at the same supersaturation?

12. Line 389. My guess is that the CCN activation ratio was higher because the particles were larger. I doubt if it has anything to do with the 3 conditions you mention.

13. Line 390. This could be the explanation or it could be coagulation.

14. Line 454. Can't you say how the three DMS instruments compared?

15. Line 490. Can the comparison be quantified here?

16. Line 510. What was the result?

17. Line 576. Influence of SSM on air-sea exchange?

18. Line 579. Entrainment. Can you say more about this in the manuscript?

19. Line 584. Chl-a is an indicator of plankton biomass, not productivity.

20. Line 602. Where are the rest of the data available?

21. Figure 2. What is the line?

---

## Referee Comment (RC3) · Anonymous Referee #3 · 9 Aug 2017

The manuscript involves an overview of the SOAP campaign which involves identifying relationships between biogeochemistry and marine boundary layer aerosol in the remote ocean. Several measurements were conducted in biologically productive waters east of New Zealand. The introduction of the manuscript is well written and provides a detailed background of previous work on the subject. The remainder of the manuscript thoroughly describes the measurements made throughout the campaign. While the manuscript is well written, only basic results comparing measurements to previous campaigns or models, are mentioned. It is mostly an overview of the measurements made and not the results or analysis. I realize the manuscript is an overview paper and the authors can not include all the results of other manuscripts that are in the works, but I was expecting a bit more analysis or at least some key findings followed by a

citation to another SOAP manuscript where I can learn more. Based on the current manuscript, I do not know what manuscripts or analysis I should look forward to. Overall, I suggest accepting the manuscript after minor revisions. I encourage the authors to clearly identify some key findings.

minor comments: Pg. 11. I may have missed it, but were particles dried before they were measured?

line 390- is there analysis behind this or is this speculation based on the marine conditions?

---

## Author Comment (AC1) · 18 Sep 2017

The SOAP experiment is one of the largest, most comprehensive and most interesting efforts conducted so far to study biogeochemical surface ocean – lower atmosphere interactions. Some articles on topical studies within the global study have been or are being published, but there is the critical need for an overview paper like this that provides the context and describes the experimental approach. The present manuscript is definitely worth publishing to serve this purpose, even though it falls a bit short in enunciating the main findings and advances of knowledge. I particularly like the introduction, which does a very good job with summarizing the state of the art, the gaps of knowledge and the need for such an experiment. The oceanographic and environmental regional context is very much appreciated too. The other aspect I like best is

the listing of the instruments and how they complement one another. This is something typically missing in many papers for a lack of space, and that the nature of this manuscript allows.

Thank you for these comments. We acknowledge that the paper does not include all the main findings, as some are still being evaluated and/or are not published. However, it does incorporate all published results to date, and also preliminary insights into some unpublished data, as well a revision of the regional mean DMSsw. However, to reflect that the paper is not a comprehensive report on all the SOAP results we have adjusted the title to: "An Overview and Preliminary Results of the Surface Ocean Aerosol Production (SOAP) campaign"

I miss comparison with previous similar cruises, such as the ACSOE or the SAGE, and statement of what is different and how SOAP goes a step forward.

We have included a Supplementary Table that briefly summarises previous campaigns, and added this text to the Introduction: " Previous related research campaigns have examined the biogeochemical and physical factors influencing oceanic DMS and $CO_2$ň fluxes, as summarised in Suppl. Table 1, but few have linked this to the physical controls of air-sea exchange, and variation in aerosol and trace gas composition of the MBL. Similarly, other campaigns with an atmospheric focus, such as MAP (Decesari et al., 2011), have carried out detailed studies of aerosol chemistry, but not interpreted this with regard to surface ocean biogeochemistry. To address this.... " and also: "Building upon the approaches used in previous studies, the SOAP campaign targeted three phytoplankton blooms of differing plankton community composition, to determine their respective influences on biogeochemistry, gas exchange and MBL composition"

In terms of a bloom-related study, SOAP is a bit disappointing. I mean, the links between each of the blooms, its biogeochemical processes, and the results of the air-sea exchange, are weak. Effort is made in the present manuscript to argument that each of the situations or blooms is not a static environment but dynamic, with changes associated with meteo forcing and so forth. This is sharp and honest – the drawback is that the blooms were not very clearly delineated so that process-based associations with aerosol precursors of more general applicability could be built. Do you the authors agree with this analysis? Along these same lines, the recent paper by Royer et al. (2016) in Scientific Reports shows dramatic changes in DMS concentration associated with the passage of a storm.

Although the SOAP Overview paper does not include direct evidence of links between bloom biogeochemistry & air-sea exchange in the figures, it summarises the results of SOAP publications that do address this. For example, Bell et al (2015) directly link surface DMS distribution to DMS flux, by calculating a flux footprint and highlighting the importance of considering the dynamics of the marine source. In addition, Walker et al (2017) relate near-surface DMS distribution and biogeochemistry to EC-derived DMS fluxes and kDMS. In addition, there will be forthcoming papers that relate bloom biogeochemistry to DMS and $CO_2$ flux, and aerosol composition.

Specifics -Line 207: Mahajan et al. 2006 should read 2015

Corrected in text and references

-Line 314: remove parenthesis after 9nmol L-1

Done

-Page 17: when discussing about the underestimation of the current climatology for the region, and call for a revision into much higher concentration, to what extent do you think your numbers are biased high because you deliberately visited blooms? What can you say about average regional concentrations?

We have addressed this by adding the following to the Discussion: "Although the Pre-SOAP and SOAP sampling strategy of focussing on phytoplankton blooms may introduce bias towards higher DMSsw, the BOX voyage, which had broad spatial coverage of subtropical and subantarctic waters between 39.5-47oS, gave a similar mean

DMSsw to the weighted mean for all voyages."

And also changed the wording in the Abstract to: "Inclusion of SOAP data in a regional DMSsw compilation indicates that the current climatological mean is an underestimate"

-Page 19: To me, it is pretty obvious that instantaneous correlations between chla and the aqueous concentration of DMS or any other biogenic volatile can be expected (yet not always found),......

We have added additional information here: "There was a weak, but significant correlation (r = 0.12, p< 0.005) between Chl-a and DMSsw in the underway surface data during SOAP, but also significant variability in the slope and the sign of this relationship between the different blooms."

…...but not necessarily with the flux. The flux depends primarily on the aqueous concentration but also on e.g. the wind speed. Therefore, correlations between biological markers and the emission flux are to be expected, if anything, over longer time scales.

We agree, but have retained this observation with a caveat added: "ÂňCorrelations were apparent during SOAP between Chl-a and DMSP (Lizotte et al., submitted), and Chl-a with DMSsw and DMSa, but there was no relationship between Chl-a and DMS flux, as expected, due to the short timescales and flux footprint identified by Bell et al., 2015."

Please also note the supplement to this comment:
https://www.atmos-chem-phys-discuss.net/acp-2017-535/acp-2017-535-AC1-supplement.pdf

**Supplement:**

**Supplementary Information**

| Voyage | Location | Date | Science focus | Observational framework | Summary publication |
|---|---|---|---|---|---|
| ACE-1 | Sub-Antarctic 40-58°S 135-157°E | Nov-Dec 1995 | Aerosol processes and production in the remote MBL | Shipboard biogeochemistry & processes coordinated, ground stations and over flight atmospheric sampling | Griffiths et al (1999) |
| GASEX - 98 | Atlantic | May-June 1998 | Direct & indirect estimates of KCO2, and associated physical drivers & biogeochemistry | Lagrangian dual tracer study in algal bloom in warm core eddy | McGillis et al (2001) |
| ACSOE | N. Atlantic | June-July 1998 | Trace gases, biogeochemistry and productivity | Lagrangian $SF_6$ tracer in anti-cyclonic eddy | Jickells et al (2008) |
| DISCO | N. Sea | June 1999 | Processes, rates and controls on marine DMS cycling | Lagrangian $SF_6$ tracer in a coccolithophore bloom | Burkill et al (2002) |
| SAGE | S. W. Pacific sub-Antarctic | March-April 2004 | The influence of iron addition on trace gas cycling and biogeochemistry, with estimates of K and physical processes | Lagrangian dual tracer study with iron addition in cyclonic eddy | Harvey et al (2010) |
| P2P | Offshore, Tasmania | 2006 | Secondary organic aerosol, MBL nucleation, coastal macroalgae, iodine | Cape Grim station based with offshore coastal vessel measurements | Cainey et al (2007) |
| DOGEE-SOLAS | N. Atlantic | June-July 2007 | Parameterizing the gas exchange by constraining K and its physical controls | Artificial surfactant release in 3 Lagrangian dual tracer experiments | Salter et al (2011) |
| GASEX III | Southern Ocean | Feb-April 2008 | K and air-sea CO2 flux estimation, including physical controls and some biogeochemistry | 2 Lagrangian dual tracer studies | Ho et al (2011) |
| ACSOS | Arctic | Aug-Sept 2008 | Formation and life cycle of low-level Arctic clouds; the interaction with the sea ice and ocean and associated physical, chemical, and biological processes | Repeat regional shipborne transects, complemented by on-ice & airborne measurements | Tjernström et al (2014) |
| VAMOS | S.E. Pacific | Oct-Nov 2008 | Determine links between aerosols, clouds and precipitation and impacts on stratocumulus radiative properties, and the physical and chemical couplings between the upper ocean and the lower atmosphere | Airborne & shipborne platforms complemented by fixed coastal sites & moorings | Wood et al (2011) |

| SIPEX(II) | Australian Antarctic Sector | Sep-Nov 2011 | Biogeochemical processes and aerosol associated with sea-ice | East Antarctic pack ice (south of 61.5 S, between 112 and 122 E) | Vancoppenolle et al 2013; Humphries et al 2016 |
|---|---|---|---|---|---|
| SOAP | S. W. Pacific Frontal waters | Feb 2011 & March-April 2012 | Determine the physical and biogeochemical controls of k, DMS and $CO_2$ fluxes, and also on aerosol & MBL composition | Lagrangian drifter study of 3 different phytoplankton blooms | This paper |

**Table S1. Selected related multidisciplinary studies of air-sea interaction related to the SOAP campaign.**

**References**

Burkill PH, Archer SD, Robinson C, Nightingale PD, Groom SB, Tarran GA, Zubkov MV. Dimethyl sulphide biogeochemistry within a coccolithophore bloom (DISCO): an overview. *Deep Sea Research Part II: Topical Studies in Oceanography* 49(15): 2863-2885. 2002.

Cainey JM, Keywood M, Grose MR, Krummel P, Galbally IE, Johnston P, Gillett RW, Meyer M, Fraser P, Steele P, Harvey M, Kreher K, Stein T, Ibrahim O, Ristovski ZD, Johnson G, Fletcher CA, Bigg EK, Gras JL. Precursors to Particles (P2P) at Cape Grim 2006: Campaign Overview, Environmental Chemistry, 4:143-150, doi:10.1071/EN07041, 2007.

Griffiths FB, Bates TS, Quinn PK, Clementson LA, Parslow JS. Oceanographic context of the First Aerosol Characterization Experiment (ACE 1): A physical, chemical, and biological overview. *Journal of Geophysical Research: Atmospheres*, *104*(D17): 21649-21671. 1999.

Harvey MJ, Law CS, Smith MJ, Hall JA, Abraham ER, Stevens C, Hadfield M, Ho DT, Ward B, Archer SD, Cainey J, Currie K, Devries D, Ellwood M, Hill P, Jones GB, Katz D, Kuparinen J, Macaskill B, Main W, Marriner A, McGregor J, McNeil C, Minnett PJ, Nodder S, Peloquin J, Pickmere S, Pinkerton M, Safi K, Thompson R, Walkington M, Wright SW, Ziolkowski L. The SOLAS Air-Sea Gas Exchange Experiment (SAGE) 2004. *Deep-Sea Res II,* 58:753-763. 2011.

Ho DT, Sabine CL, Hebert D, Ullman DS, Wanninkhof R, Hamme RC, Strutton PG, Hales B, Edson JB, Hargreaves BR.  Southern Ocean Gas Exchange Experiment: Setting the Stage, *J. Geophys. Res.*, 116, C00F08, doi:10.1029/2010JC006852. 2011.

Humphries RS, Klekociuk, AR, Schofield, R, Keywood, M, Ward, J, and Wilson, SR. Unexpectedly high ultrafine aerosol concentrations above East Antarctic sea ice, AtmosChemPhys, 16, 2185-2206, doi:105194/acp-16-2185-2016. 2016.

Jickells TD, Liss PS, Broadgate W, Turner S, Kettle AJ, Read J, Baker J, Cardenas LM, Carse F, Hamren-Larssen M, Spokes L, Steinke M, Thompson A, Watson AJ, Archer SD, Bellerby RGJ, Law CS, Nightingale PD, Liddicoat MI, Widdicombe CE, Bowie A, Gilpin LC, Moncoiffé G, Savidge G, Preston T, Hadziabdic P, Frost T, Upstill-Goddard RC, Pedrós-Alió C, Simó R, Jackson A, Allen A, DeGrandpre. Lagrangian biogeochemical study of an eddy in the North East Atlantic, *Prog. Oceanography* 76:366–398. 2008.

McGillis WR, Edson JB, Ware JD, Dacey JW, Hare JE, Fairall CW, Wanninkhof R. Carbon dioxide flux techniques performed during GasEx-98. Marine Chemistry 75(4):267-80. 2001.

Salter ME, Upstill-Goddard RC, Nightingale PD, Archer SD, Blomquist B, Ho DT, Huebert B, Schlosser P, Yang M. Impact of an artificial surfactant release on air-sea gas fluxes during Deep Ocean Gas Exchange Experiment II. *Journal of Geophysical Research: Oceans* 116, C11. 2011

Tjernström M, Leck C, Birch CE, Bottenheim JW, Brooks BJ, Brooks IM, Bäcklin L, Chang RW, de Leeuw G, Di Liberto L, De La Rosa S. The Arctic Summer Cloud Ocean Study (ASCOS): overview and experimental design. Atmospheric Chemistry and Physics. 14(6):2823-69. 2014.

Wood R, Mechoso CR, Bretherton CS, Weller RA, Huebert B, Straneo F, Albrecht BA, Coe H, Allen G, Vaughan G, Daum P. The VAMOS ocean-cloud-atmosphere-land study regional experiment (VOCALS-REx): goals, platforms, and field operations. Atmospheric Chemistry and Physics 11(2):627-54. 2011.

Vancoppenolle M, Meiners KM, Michel C, Bopp L, Brabant F, Carnat G, Delille B, Lannuzel D, Madec G, Moreau S, Tison J-L, van der Merwe, P. Role of sea ice in global biogeochemical cycles: emerging views and challenges, Quaternary Science Reviews, 79:207-230. 2013.

| Station | NZST Date | NZST Time | Latitude | Longitude | SST (°C) | Salinity | MLD (m) | Chl-a (mg m$^{-3}$) | Nitrate (umol L$^{-1}$) | Location |
|---|---|---|---|---|---|---|---|---|---|---|
| 7502 | 15-Feb-12 | 9:00 | -44.608 | 174.773 | 13.92 | 34.43 | 14.2 | 1.02 | 5.91 | B1 |
| 7503 | 16-Feb-12 | 9:28 | -44.583 | 174.700 | 14.44 | 34.49 | 13.33 | 0.68 | 5.56 | B1 |
| 7504 | 17-Feb-12 | 9:14 | -44.550 | 174.712 | 14.99 | 34.56 | 14.35 | 1.08 | 4.02 | B1 |
| 7505 | 18-Feb-12 | 9:19 | -44.574 | 174.736 | n/a | n/a | 14.35 | 0.52 | 0.11 | B1 |
| 7506 | 19-Feb-12 | 7:34 | -44.336 | 175.243 | 14.68 | 34.44 | 16 | 0.88 | 5.86 | North of B1 |
| 7507 | 20-Feb-12 | 7:21 | -45.960 | 173.645 | 13.58 | 34.40 | 22.94 | 0.74 | 3.77 | S.W. station |
| 7508 | 21-Feb-12 | 7:02 | -43.741 | 176.965 | 15.12 | 34.61 | 22.95 | 0.65 | 1.26 | Enroute to E |
| 7509 | 21-Feb-12 | 17:20 | -43.483 | 179.114 | 15.88 | 34.77 | 21.8 | 0.28 | <0.07 | East of B2 |
| 7510 | 22-Feb-12 | 9:22 | -43.717 | 180.157 | 15.88 | 34.65 | 21.12 | 1.39 | 0.08 | B2 |
| 7511 | 22-Feb-12 | 13:28 | -43.597 | 180.179 | 15.90 | 34.66 | 21.12 | n/a | n/a | B2 |
| 7512 | 22-Feb-12 | 18:01 | -43.627 | 180.208 | 16.34 | 34.64 | 14.25 | n/a | n/a | B2 |
| 7513 | 23-Feb-12 | 7:17 | -43.710 | 180.238 | 15.72 | 34.56 | 18.96 | 0.58 | 2.62 | B2 |
| 7514 | 23-Feb-12 | 9:43 | -43.699 | 180.228 | 15.65 | 34.55 | 18.24 | 0.55 | 2.50 | B2 |
| 7515 | 23-Feb-12 | 12:06 | -43.691 | 180.257 | 15.92 | 34.64 | 24.91 | 0.64 | 0.47 | B2 |
| 7516 | 23-Feb-12 | 17:14 | -43.641 | 180.247 | 15.53 | 34.60 | 38.89 | 0.53 | 1.56 | B2 |
| 7517 | 24-Feb-12 | 7:03 | -43.667 | 180.236 | 15.56 | 34.58 | 38.89 | 0.61 | 1.32 | B2 |
| 7518 | 24-Feb-12 | 15:17 | -43.601 | 180.235 | 15.73 | 34.65 | 11.23 | 0.48 | 1.15 | B2 |
| 7519 | 25-Feb-12 | 9:26 | -43.557 | 180.316 | 15.44 | 34.56 | 29.23 | 0.67 | 2.86 | B2 |
| 7520 | 25-Feb-12 | 14:31 | -43.630 | 180.260 | 15.59 | 34.55 | 26.92 | 0.63 | 2.92 | B2 |
| 7521 | 26-Feb-12 | 6:55 | -43.963 | 180.692 | 15.77 | 34.78 | 25 | 0.51 | 0.13 | South of B2 |
| 7522 | 27-Feb-12 | 15:28 | -44.112 | 175.475 | 14.70 | 34.51 | 28.93 | 0.44 | 3.94 | NE B3a |
| 7523 | 28-Feb-12 | 7:25 | -44.491 | 174.850 | 14.24 | 34.53 | 30.9 | 0.39 | 2.21 | B3 a |
| 7524 | 28-Feb-12 | 13:10 | -44.542 | 174.873 | 14.36 | 34.49 | 29.46 | 0.29 | 4.40 | B3 a |
| 7525 | 29-Feb-12 | 8:52 | -44.607 | 174.870 | 14.19 | 34.49 | 27.03 | 0.44 | 4.42 | B3 a |
| 7526 | 29-Feb-12 | 13:03 | -44.545 | 174.883 | 14.39 | 34.50 | 27.03 | n/a | n/a | B3 a |
| | | | | | | | | | | Storm |
| 7527 | 2-Mar-12 | 7:58 | -44.191 | 174.925 | 13.62 | 34.52 | 39.07 | 0.42 | 5.28 | B3 b |
| 7528 | 2-Mar-12 | 14:57 | -44.192 | 174.927 | 13.71 | 34.54 | 44.79 | 0.45 | 2.93 | B3 b |
| 7529 | 3-Mar-12 | 7:42 | -44.759 | 174.640 | 13.27 | 34.38 | 30.72 | 0.61 | 5.34 | B3 b |
| 7530 | 3-Mar-12 | 14:45 | -44.781 | 174.652 | 13.03 | 34.39 | 44.11 | 0.49 | 5.49 | B3 b |
| 7531 | 4-Mar-12 | 9:26 | -44.243 | 174.523 | 12.88 | 34.50 | 49.55 | 0.74 | 3.86 | B3 b |
| 7532 | 4-Mar-12 | 15:23 | -44.243 | 174.522 | 12.83 | 34.48 | 39.51 | 1.01 | 3.81 | B3 b |
| 7533 | 5-Mar-12 | 9:57 | -44.191 | 174.307 | 12.79 | 34.50 | 40.14 | 0.97 | 2.84 | B3 b |

Supplementary Table 2. SOAP CTD station information, including surface water characteristics. Individual bloom occupation is indicated by the different colours. SST: Sea Surface Temperature; MLD: Surface Mixed layer depth; Chl-*a*: Chlorophyll-*a*.

---

## Author Comment (AC2) · 18 Sep 2017

This manuscript provides an overview of the multi-disciplinary SOAP cruise off the coast of New Zealand in 2012. I believe such an overview is important and that the manuscript should be published with the following modifications:

Thanks for these comments

1. Line 37 You don't show a correlation between chlorophyll-a and DMSsw.

We do not show the correlation in a figure, but have added the following to the Conclusions: "Overall there was an weak, but significant, correlation ($r = 0.12$, $p < 0.005$) between Chl-a and DMSsw in the underway surface data during SOAP, but also significant variability in the slope and the sign of this relationship between the different

blooms"

3. Line 80 You are mixing aerosol mass and number here.

Rewritten to improve clarity: "Breaking waves and associated bubble formation are a major source of Primary Marine Aerosol (PMA), supplying most the aerosol mass in the marine boundary layer (MBL) over the remote ocean (Andreae and Rosenfeld, 2008), and particularly in regions that experience high winds and breaking waves (de Leeuw et al., 2014). This is reflected in PMA contributing only ∼10–20% of CCN number concentrations over the remote Pacific Ocean (Blot et al. 2013; Clarke et al. 2013), but up to 55% over the Southern Ocean (McCoy et al. 2015)."

4. Line 173 What is secondary production?

Biomass production by consumers (as opposed to primary production by phytoplankton)

5. Line 199 Should read "aerosols and their precursors".

Changed to "aerosols and precursors"

6. Line 263 Could you please give more details on the bubble chamber.

Now added: "The composition of primary marine aerosols was also examined using a 0.45m3 bubble chamber, in which sea spray was formed via the bursting of bubbles produced by passing clean compressed air through sintered glass (Mallet et al., 2016).

7. Line 263. The Supplementary table should be in the main manuscript. It would be helpful to have a reference for each measurement.

This table is now in the main manuscript as Table 1, but we have not added references for each measurement, as this would require too many additional references.

7. Line 276. What do you mean by "biogeochemical signals"?

Modified to "elevated chl-a and DMSsw, and pCO2 drawdown"
8. Figure 5 needs to be larger to make it more readable.

Fig. 5 is now revised, so that text and labels are clearly visible

9. What is the light blue line in Figure 6b?

The cyan line indicates wind direction. This was noted in the Figure legends in the text, but omitted from the legend below Figure 6

10. Line 340. Aerosol number concentration:

Corrected to "Aerosol number concentration"

11. Line 386. CCN data should include the % supersaturation. Were all measurements made at the same supersaturation?

Now added "at 0.5% supersaturation"

12. Line 389. My guess is that the CCN activation ratio was higher because the particles were larger. I doubt if it has anything to do with the 3 conditions you mention.

We have added "the median particle diameters during clean marine periods were consistent between the three blooms" which contradicts the referees' suggestion, and so retained the comment that "that particle composition, secondary organics or coagulation may have impacted CCN activation at B1". This is also further supported by: "preliminary results from an application of the ACCESS-UKCA model (Woodhouse, pers comm.), which simulated the additional impact of emissions of marine secondary organic carbon under the conditions determined during SOAP.

13. Line 390. This could be the explanation or it could be coagulation.

Coagulation is now included as an alternative reason for particle growth (see above)

14. Line 454. Can't you say how the three DMS instruments compared?

Now added: "Intercomparison of the PTR-MS and SCD during SOAP, involved analysis of two air samples and two diluted DMS gas standards with a concentration range of

158 – 354 ppt. The instruments showed very good agreement, with a mean difference of 5% and maximum 10%."

15. Line 490. Can the comparison be quantified here?

The comparison of the micrometeorological techniques is in Smith et al. (to be submitted).

16. Line 510. What was the result?

Now expanded to: "In addition, SOAP data was used to parameterise whitecap coverage against wind-speed, and identify that maturing waves may obscure and lead to underestimate of the variability of breaking waves (Scanlon and Ward, 2016).

17. Line 576. Influence of SSM on air-sea exchange?

Now expanded to: "on DMS emissions"

18. Line 579. Entrainment. Can you say more about this in the manuscript?

The sentence mentioning entrainment has been removed

19. Line 584. Chl-a is an indicator of plankton biomass, not productivity.

Changed to "phytoplankton biomass"

20. Line 602. Where are the rest of the data available?

Added: "The remaining data is available by request email to cliff.law@niwa.co.nz"

21. Figure 2. What is the line?

The line has been removed from Figure 2

---

## Author Comment (AC3) · 18 Sep 2017

The manuscript involves an overview of the SOAP campaign which involves identifying relationships between biogeochemistry and marine boundary layer aerosol in the remote ocean. Several measurements were conducted in biologically productive waters east of New Zealand. The introduction of the manuscript is well written and provides a detailed background of previous work on the subject. The remainder of the manuscript thoroughly describes the measurements made throughout the campaign. While the manuscript is well written, only basic results comparing measurements to previous campaigns or models, are mentioned. It is mostly an overview of the measurements made and not the results or analysis. I realize the manuscript is an overview paper and the authors can not include all the results of other manuscripts that are in the works,

but I was expecting a bit more analysis or at least some key findings followed by a citation to another SOAP manuscript where I can learn more. Based on the current manuscript, I do not know what manuscripts or analysis I should look forward to. Overall, I suggest accepting the manuscript after minor revisions. I encourage the authors to clearly identify some key findings.

Thank you for these comments. We acknowledge that we do not identify all the findings of the SOAP campaign, although we do include the results from all SOAP papers published to date and cite these accordingly. In addition, we include some unpublished SOAP results as pers. comm. or submitted, with 9 published and submitted papers cited that discuss SOAP campaign results. We also present new data, by using from a number of research voyages in the South Pacific to refine the regional DMSsw mean. Nevertheless, to address this referees', and also Ref 1's comments, we have adjusted the title of the paper to reflect that this overview only contains preliminary results ("An Overview and Preliminary Results of the Surface Ocean Aerosol Production (SOAP) campaign").

Pg. 11. I may have missed it, but were particles dried before they were measured?

We have now clarified this in the figure 9 legend: "Ambient RH measurement was used for RH correction of the PCASP, Hi Vol and SMPS, and diffusion driers (Silica Gel) were used on the inlet of the UFO-TDMA and VH-TDMA."

Line 390 - is there analysis behind this or is this speculation based on the marine conditions?

This is now supported by the following additional information: "These findings are supported by preliminary modelling results with the ACCESS-UKCA model which has been used to simulate the additional impact of emissions of marine secondary organic carbon, with respect to observations from SOAP on the local scale (Woodhouse, pers comm.)"